

**Interaction between Dicarboxylic Acid and Sulfuric Acid-Base Clusters**
**Enhances New Particle Formation**
Yun Lin,[1] Yuemeng Ji,[1,2,*] Yixin Li,[1] Jeremiah Secrest,[1] Wen Xu,[3] Fei Xu,[4] Yuan Wang,[5]
Taicheng An,[2] and Renyi Zhang[1,*]
[1]Department of Atmospheric Sciences and Department of Chemistry, Texas
A&M University, College Station, TX 77843, USA
[2]Institute of Environmental Health and Pollution Control, Guangdong University of Technology,
Guangzhou 510006, China
[3]Aerodyne Research Inc, Billerica, MA 01821, USA
[4]School of Environmental Science and Engineering, Shandong University, Jinan 250100, China
[5]Division of Geological and Planetary Sciences, California Institute of Technology, Pasadena,
CA, 91125, USA
* Corresponding authors: Renyi Zhang, renyi-zhang@tamu.edu; Yuemeng Ji, jiym99@163.com





**ABSTRACT.** Dicarboxylic acids are believed to stabilize pre-nucleation clusters and facilitate
new particle formation in the atmosphere, but the detailed mechanism leading to the formation of
multi-component critical nucleus involving organic acids, sulfuric acid (SA), base species, and
water remains unclear. In this study, theoretical calculations are performed to elucidate the
interactions between succinic acid (SUA) and clusters consisting of SA-ammonia
(AM)/dimethylamine (DMA) in the presence of hydration of up to six water molecules. Formation
of the hydrated SUA·SA·base clusters by adding one SUA molecule to the SA·base hydrates is
energetically favorable. The addition of SUA to the SA·base hydrates either triggers proton
transfer from SA to the base molecule, resulting in formation of new covalent bonds, or strengthens
the pre-existing covalent bonds. The presence of SUA promotes hydration of the SA·AM and
SA·AM·DMA clusters but dehydration of the SA·DMA clusters. At equilibrium, the uptake of
SUA competes with the uptake of the second SA molecule to stabilize the SA·base clusters at
atmospherically relevant concentrations. The clusters containing both the base and organic acid
are capable of further binding with acid molecules to promote their subsequent growth. Our results
indicate that the multi-component nucleation involving organic acids, sulfuric acid, and base
species promotes new particle formation in the atmosphere, particularly under polluted conditions.





## 1.    NTRODUCTION

Atmospheric aerosols are important to several issues, including climate, visibility and
human health (IPCC, 2013; Zhang et al., 2015). In particular, aerosols influence the Earth energy
budget directly by absorbing/scattering incoming solar radiation and indirectly by acting as cloud
condensation nuclei (CCN)/ice nuclei (IN), which impact the lifetime, coverage, precipitation
efficiency, and albedo of clouds (Andreae et al., 2004; Fan et al., 2007; Li et al., 2008). Currently,
the indirect radiative forcing of aerosols represents the largest uncertainty in climate predictions
(IPCC, 2013). New particle formation (NPF) has been observed under diverse environmental
conditions (Kulmala and Kerminen, 2008; Zhang et al., 2012; Guo et al., 2014; Bianchi et al., 2016;
Wang et al., 2016) and contributes up to half of the CCN population in the troposphere (Merikanto
et al., 2009; Yue et al., 2011). NPF involves two distinct steps, i.e., nucleation to form a critical
nucleus and subsequent growth of newly formed nanoparticles to a larger size (> 3 nm). Currently,
the identities and the roles of chemical species involved in NPF are not fully understood at the
molecular level, hindering the development of physically-based parameterization to include NPF
in atmospheric models (Zhang et al. 2010; Cai and Jiang, 2017). Sulfuric acid (SA) is believed to
be the most common atmospheric nucleation species, and ammonia (AM)/amines further stabilize
the hydrated sulfuric acid clusters and enhance the nucleation (Kuang et al, 2010; Yue et al., 2010;
Erupe et al., 2011; Yu et al., 2012; Qiu et al., 2013; Wang et al., 2018; Yao et al., 2018). However,
neither the sulfuric acid-water binary nucleation nor ammonia/amine-containing ternary
nucleation sufficiently explains the measured NPF rates in the lower troposphere (Xu et al., 2010a;
Zhang et al. 2012), suggesting the role of other chemical species, such as organic acids, in NPF
(Zhang et al., 2004; McGraw and Zhang, 2008).



The role of organic species in assisting aerosol nucleation and growth has been
demonstrated by both experimental and theoretical studies (Zhang et al., 2009; Zhao et al., 2009;
Wang et al., 2010; Xu et al., 2010b; Xu et al., 2012; Elm et al., 2014; Weber et al., 2014; Xu et al.,
2014; Zhu et al., 2014; Tröstl et al., 2016). However, the interactions between organic acids and
the other nucleation precursors are still elusive, due to the large variability in the physicochemical
nature of organic acids, e.g., the wide range of volatility and functionality (Zhang et al., 2012;
Riccobono et al., 2014). In addition, most of the previous theoretical studies focus on the
enhancement effects of organic acids on the SA-$H_2O$ binary nucleation or the role of organic acids
in clustering basic species such as ammonia or amines with hydration (Zhao et al., 2009; Xu et al.,
2010; Xu et al. 2013; Elm et al., 2014; Weber et al., 2014; Zhu et al., 2014). Several recent studies
have been conducted on the underlying mechanisms of organic acids in large pre-nucleation
clusters (e.g. ammonia/amine-containing ternary nucleation) (Xu et al., 2010; Xu et al., 2012; Elm
et al., 2016; Zhang et al., 2017), but most of these studies treated the clusters without the
consideration of hydration. Because of the ubiquitous presence of water (W) in the atmosphere
and its much higher abundance than other nucleation precursors, the hydration effect on aerosol
nucleation is significant (Loukonen et al., 2010; Xu and Zhang, 2013; Henschel et al., 2014; Zhu
et al., 2014; Henschel et al., 2016).
Atmospheric measurements have shown that the presence of dicarboxylic acids, including
succinic acid (SUA), is prevalent in ambient particles (Kawamura and Kaplan, 1987; Decesari et
al., 2000; Legrand et al., 2005; Hsieh et al. 2007; Blower et al., 2013). The effect of dicarboxylic
acids on aerosol nucleation involving SA or base molecules has been recognized in theoretical
studies. Wen and Zhang (2012) showed that dicarboxylic acids promote aerosol nucleation with
other nucleating precursors in two directions via hydrogen bonding to the two carboxylic groups



on dicarboxylic acids, which is distinct from monocarboxylic acids. Elm et al. (2014) indicated
that clustering of a single pinic acid with SA molecules leads to closed structures because of no
available sites for additional hydrogen bonding. In addition, Elm et al. (2017) suggested that more
than two carboxylic acid groups are required for a given organic oxidation product to efficiently
stabilize sulfuric-acid contained clusters. The interaction between SUA and dimethylamine (DMA)
is strengthened by hydration via forming aminium carboxylate ion pairs (Xu and Zhang, 2013),
while hydration of oxalic acid-AM cluster is somewhat unfavorable under atmospheric conditions
(Weber et al., 2014). Clearly, the interactions of dicarboxylic acids with other nucleation
precursors depend on the type of dicarboxylic acids and the number of the molecules involved in
clustering. Presently, theoretical studies on the effect of dicarboxylic acids on nucleation from
multi-component systems are lacking (Xu et al., 2010; Xu and Zhang, 2013). In particular, the role
of organic acids as well as their participation in stabilizing larger pre-nucleation clusters of the
SA-ammonia/amine systems needs to be evaluated with the presence of hydration in order to better
understand NPF.

In this study, we performed theoretical calculations to evaluate the effect of SUA on

hydrated SA·base clusters. Two base species, ammonia and dimethylamine (DMA), were
considered. The Basin Paving Monte Carlo (BPMC) method was employed to sample stable
cluster conformers, and quantum calculations were performed to predict the thermochemical
properties of the multi-component clusters. Geometric and Topological analyses were carried out
to investigate the binding pattern between SUA and SA·base clusters in the presence of hydration.
The implications of the interaction of SUA with hydrated SA-base clusters for atmospheric NPF
are discussed.
**2.      COMPUTATIONAL METHODS**





The methodology of the BPMC conformational sampling combined with quantum
calculations using density functional theory (DFT) was employed to assess the role of SUA in
clustering of SA with base compounds in the presence of water (Xu and Zhang, 2013). Briefly, the
local energy minima in BPMC simulations was searched by using Amber11 package, and the Basin
Hopping Monte Carlo (BHMC) approach was employed to increase the Monte Carlo transition
probability, which allows the clustering system to escape from the traps of local energy minimum.
Hydration of the clustering system was evaluated by employing the TIP3P model. The geometric
optimization and frequency calculations of the BPMC sampled cluster complexes were further
performed at PW91PW91 level of theory with the basis set 6-311++G(2d, 2p) using Gaussian 09
software package (Frisch et al., 2009). Thermodynamic quantities, such as the electronic energy
($\Delta E$ with ZPE), enthalpy ($\Delta H$), and Gibbs free energies ($\Delta G$), were obtained on the basis of
unscaled density functional frequencies at temperature of 298.15 K and pressure of 1 atm. Several
basic cluster systems were also examined at the M06-2X/6-311++G(3df,3pd) level of theory,
which has been suggested to be more reliable in predicting binary/ternary cluster formation (Elm
and Mikkelsen, 2012; Leverentz et al., 2013; Zhang et al., 2017). Comparisons of the free energies
with the two different DFT levels of theories between this study and previous available theoretical
and experimental studies are presented in Table 1. The energies derived at the PW91PW91/6-
311++G(2d, 2p) are consistent with those of the M06-2X/6-311++G(3df,3pd) method, and the
differences between our calculations and previous studies are within 1.6 kcal mol[-1].
Topological analysis on the SA·base clusters with hydration and SUA was performed by
employing the Multifunctional Wavefunction Analyzer (Multiwfn) 3.3.8 program (Lu and Chen,
2012). The topological characteristics at the bond critical points (BCPs) were calculated for
electron density ($\rho$), Laplacian of electron density ($\Delta\rho$), and potential energy density ($V$). Since the





electron density is highly correlated to bonding strength (Lane et al., 2013), the potential energy
density is an indicator of hydrogen bond energies (Espinosa et al., 1998). The occurrence of proton
transfer in the clusters was determined using the localized orbital locator (LOL). A high LOL value
denotes greatly localized electrons and indicates the existence of a covalent bond (Lu et al., 2012).
The covalent bond is characterized by a negative $\Delta\rho$, while a positive $\Delta\rho$ is associated with a
hydrogen bond. In addition, a newly formed covalent bond via proton transfer was quantitatively
examined in terms of the bond strength using the Laplacian bond order (LBO) as an indicator (Lu
et al., 2013). Both LOL and LBO were calculated with Multiwfn 3.3.8 program (Lu et al., 2012).

The extent, to which clusters are hydrated (or the hydrophilicity of the clusters), is affected

by humidity conditions in the atmosphere (Loukonen et al., 2010; Henschel et al., 2014; Henschel
et al., 2016). To examine the influence of SUA on cluster hydration under different humidity
conditions, the relative hydrate distributions over the number of water molecules contained in
clusters were calculated at different relative humidity (RH) levels. The distribution was calculated
according to Henschel et al. (2014), in which the Gibbs free energies of hydration obtained from
DFT calculations are converted to equilibrium constants for the formation of the respective hydrate
by

$$K = e^{\frac{-\Delta G^0}{RT}} \qquad\qquad (1)$$

and the relative hydrate population $x_n$ of the hydrate containing $n$ water molecules is determined
by

$$x_n = \left(\frac{p(\mathrm{H_2O})}{p^0}\right)^n x_0 e^{\frac{-\Delta G_n}{RT}} \qquad\qquad (2)$$

where $p(\mathrm{H_2O})$ is the water partial pressure, $p^0$ is the reference pressure (1 atm), $x_0$ is the population
of the dry cluster chosen so that $\sum_0^6 x_n = 1$, $T$ is the temperature (298.15 K in this study), and $R$ is
the molar gas constant. $p(\mathrm{H_2O})$ is related to RH through



$$p(\mathrm{H_2O}) = p(\mathrm{H_2O})_{eq} \times RH \qquad (3)$$
where $p(\mathrm{H_2O})_{eq}$ is the water saturation vapor pressure, which is a function of the temperature
following Wexler (Wexler, 1976) Note that only the Gibbs free energy for the single lowest energy
structure for each system of hydration was considered in the calculation, since the Boltzmann
averaging effect over configurations on comparable clusters has been found to be negligible in
most cases, particularly for the free energies of hydration (Erupe et al., 2011;Xu and Zhang, 2013;
Tsona et al., 2015).

To assess the importance of uptake of SUA on the SA·base clusters relative to the uptake

of another one SA molecule under atmospheric condition, the ratio in the concentrations
SA·X·SUA to $(\mathrm{SA})_2$·X (X denotes either AM or DMA) is calculated assuming equilibrium
conditions,
$$\mathrm{SA \cdot X + SUA = SA \cdot X \cdot SUA} \qquad (4)$$
$$\mathrm{SA \cdot X + SA = (SA)_2 \cdot X} \qquad (5)$$
and the equilibrium constants $K_1$ and $K_2$ for reactions (4) and (5) are expressed as
$$K_1 = \frac{[\mathrm{SA \cdot X \cdot SUA}]}{[\mathrm{SA \cdot X}][\mathrm{SUA}]} = e^{\frac{-\Delta G_1}{RT}} \qquad (6)$$
$$K_2 = \frac{[(\mathrm{SA})_2 \cdot \mathrm{X}]}{[\mathrm{SA \cdot X}][\mathrm{SA}]} = e^{\frac{-\Delta G_2}{RT}} \qquad (7)$$
The concentration ratio between SA·X·SUA and $(\mathrm{SA})_2$·X is derived by dividing $K_1$ and $K_2$ and the
transformation,
$$\frac{[\mathrm{SA \cdot X \cdot SUA}]}{[(\mathrm{SA})_2 \cdot \mathrm{X}]} = \frac{[\mathrm{SUA}]}{[\mathrm{SA}]} e^{\frac{-\Delta(\Delta G)}{RT}} \qquad (8)$$
where $\Delta(\Delta G)$ is the difference in the Gibbs free energy between reactions (4) and (5) at 298 K.
The ambient concentration of SUA detected at Los Angeles is about $10^7$ molecules cm$^{-3}$
(Kawamura and Kaplan, 1987), and the typical concentration of organic acids in the atmosphere



is about $10^8$–$10^9$ molecules cm$^{-3}$ (Zhang et al., 2015). Also, a lower limit of concentration for
sulfuric acid in promoting NPF in the atmosphere is about $10^5$ molecules cm$^{-3}$ (Zhang et al., 2012).
Hence, the ratio of SA·X·SUA to (SA)$_2$·X was expressed on the basis of eq. (8) with the SUA/SA
ratio ranging from 1 to 10, 000, considering the lower limit of the concentration of SA and
available SUA in the atmosphere.
**3.   RESULTS AND DISCUSSION**
**3.1   STRUCTURES AND TOPOLOGY**
The most stable structures (in terms of $\Delta G$ at $T$=298.15K and $p$=1 atm) of the hydrated
SA·base clusters are shown in Figures 1-3. Addition of SUA to hydrated SA·base clusters alters
the bonding pattern and rearranges the cluster structure, affecting the free energy and stability for
the cluster formation.[15] Proton transfer with added SUA leads to a change in the bonding degree
for each precursor molecule. Figure 1a shows the absence of proton transfer in the SA·AM cluster,
consistent with the previous studies (Kurtén et al., 2006; Loukonen et al., 2010; Henschel et al.,
2014). When adding SUA to the cluster, proton transfer occurs (Figure 1b), which is confirmed by
the relocation of the LOL high value (Figure 4a). For the SA·AM cluster, a large value of LOL is
adjacent to the SA molecule, indicating that electrons attained to the hydrogen atom (H1) on the
S-O-H group are localized on the SA molecule side. In contrast, a large LOL region is located near
the nitrogen atom (N1) on the AM molecule with the addition of SUA, suggesting that electrons
are greatly localized on the AM side and proton transfer occurs. Because of the proton transfer,
the hydrogen bonding of N1-H1 is converted to a covalent bond, leading to the formation of
ammonium bisulfate. The existence of the new covalent bond is denoted by the available LBO
calculation, showing the bond order of the N1-H1 covalent bond with a value of 0.464. The
formation of the covalent bond is also confirmed by the negative sign of $\nabla\rho$ at BCP of N1-H1 bond



(Table S1). Along with the proton transfer, the electron density (potential energy density) at BCP
of the N1-H1 bond exhibits a significant increase (decrease), from 0.091 (-0.087) a.u. in the
SA·AM cluster to 0.271 (-0.424) a.u. in the SA·AM·SUA cluster. The structures of SA·AM and
SA·DMA hydrates with up to five water molecules in our calculations are consistent with Henschel
et al. (2014). The interactions between SA and AM/DMA in the presence of hydration have been
explored (DePalma et al., 2012; Yu et al., 2012; Qiu et al., 2013; Xu and Zhang, 2013; Tsona et
al., 2015), showing strong bonding among SA, base compound, and water molecules. Another
recent study on glycolic acid found that addition of one glycolic acid molecule to the SA·AM
cluster does not result in proton transfer, unless a second glycolic acid molecule is added (Zhang
et al., 2017). The different behavior in the SA·AM interaction between glycolic acid and SUA is
attributed to the different functional groups in the two organic acids. Hence, SUA is more efficient
than glycolic acid to stabilize the SA·AM clusters.
In contrast to the SA·AM cluster, proton transfer occurs for the SA·DMA cluster without
water or SUA (Figures 2 and 4b), because of stronger basicity of DMA than AM (Anderson et al.,
2008). Similarly, proton transfer occurs for the SA·AM·DMA cluster, leading to the formation of
the aminium bisulfate ($HSO_4^-$) ion pair. Addition of SUA to the SA·DMA·AM·$(W)_n$ system results
in additional proton transfer between the bisulfate ion and the AM molecule, leading to the
formation of sulfate double-ions ($SO_4^{2-}$) (Figures 3 and 4c).
The dependence of the number of proton transfers on the hydration level is summarized in
Table 3. The hydrations of SA clusters by Xu and Zhang (2013), who employed a similar method
for the structure sampling and quantum calculations, are also included for comparison. It is evident
from Table 3 that both hydration and addition of SUA promote the proton transfer in the SA·base
clusters. Previous studies have identified facile proton transfer by hydration (Ding et al., 2003; Al



Natsheh et al., 2004; Loukonen et al., 2010; Xu and Zhang, 2013), and the dependence of proton
transfer on the hydration level has also been suggested by Tsona et al. (2015). For the SA cluster
system, Xu and Zhang (2013) found that proton transfer in the hydrated SA clusters only occurs
with more than two water molecules. In our study, proton transfer occurs when the SA·AM and
the SA·AM·DMA·(W)$_5$ clusters are hydrated with one more water molecule (Figures 1a and 3a).
The formation of the covalent bond in the monohydrate of SA·AM and the sixth hydrate of
SA·AM·DMA is confirmed by the LOL relocation in Figure S1. Loukonen et al. (2010) also found
that proton transfer occurs in the SA·AM system for the hydrated cluster with two water molecules.
Our results show that neither water molecules nor SUA induce the second proton transfer in
SA·DMA clusters. In contrast, Loukonon et al. (2010) showed that the second proton transfer
occurs when the SA·DMA cluster is hydrated with five water molecules. The different behaviors
for proton transfer with hydration between this study and Loukonen et al. are attributable to the
different sampling methodology used to obtain the most stable conformers of the clusters. Note
that the findings of Loukonen et al. are also in contrast to those by Henschel et al. (2014)

Table 4 summarizes the available LBO values for the covalent bonds in the SA·base

clusters with hydration. The dependence of LBO on the cluster hydration level varies with the
clustering systems containing different components. For SA·AM system without SUA, additional
water molecules result in higher LBO of N1-H1 bonds, while for SA·DMA LBO of the N2-H2
bond generally increases at all hydration levels except for the dihydrate. With addition of SUA to
the SA·AM and SA·DMA systems, the LBO values of the pre-existing covalent bonds of SUA-
contained clusters are higher than those of the clusters without SUA at all hydration levels except
for the sixth hydration. This indicates that, although addition of SUA to the two hydrated cluster
systems does not result in additional proton transfer, the presence of SUA enhances the strength



of the covalent bonds formed for the initial hydration. Consistently, the electron densities (the
potential energy densities) at BCPs of the N-H bonds are somewhat higher (lower) in the SUA-
containing clusters than in those without SUA for most hydration cases (Tables S2 and S3).
Addition of SUA to the SA·base cluster results in cleavage of the original strong hydrogen
bond between the base and SA molecules (Figures 1b, 2b and 3b). As the number of water
molecules increases, the number of possible bonds among the molecules increases, leading to
complicated structures. Note that the carbon chain of SUA tends to bend accordingly as the
hydration degree increases, because both carboxylic groups at the two ends of the carbon chain in
SUA are involved in hydrogen bonding. While the bending of the carbon chain facilitates hydrogen
bonding and stabilizes the clusters, such bending also induces steric hindrance, which partially
cancels out the energy due to additional hydrogen bonding. As expected, the number of hydrogen
bonds attained by the AM molecule in SA·AM·SUA clusters increases with the hydration degree,
which is always equal or larger than that of the corresponding clusters without SUA. The number
of hydrogen bonds formed on AM molecule is closely related to the values of free energy changes
induced by addition of SUA to SA·AM clusters (see detailed discussions in following section on
the energetics). For all DMA-containing clusters, the nitrogen atom of DMA is saturated by two
hydrogen bonds, but the steric hindrance of DMA due to the two free methyl groups likely
corresponds to an important factor that affects the stability of DMA-containing clusters (Ortega et
al., 2012). The complexity of the cluster structures is partially ascribed to the formation of
intramolecular hydrogen bonding in SUA, illustrated by the clusters of SA·AM·SUA·W,
SA·DMA·SUA·W, or SA·AM·DMA·SUA·(W)$_5$ (Figures 1b, 2b and 3b).



## 3.2 ENERGETICS

The thermochemical quantities calculated at the PW91PW91/6-311++G(2d, 2p) level of theory for the most stable cluster configurations are summarized in Table 2. The free energies of hydration for the clusters, along with the number of water molecules contained in the clusters, are presented in Figure 5a. The stepwise hydration energies are provided in Table 2. For the hydration of SA·AM and SA·DMA with up to five water molecules, the hydration free energies calculated at the PW91PW91/6-311++G(2d, 2p) level are in agreement with those by Henschel et al. (2014) using the RICC2/aug-cc-pV(T+d)Z level for sulfur and the RICC2/aug-cc-pVTZ level for all other atoms, but differ from those by Loukonen et al. (2010) at the RI-MP2/aug-cc-pV(T+d)Z level of theory. Note that some of the structures of the hydrates in this study and the study of Henschel et al. are different from those of Loukonen et al. (2010), explaining the differences in the energies among the various studies.

Figure 5a shows that addition of one more water molecule to the cluster systems results in a negative hydration energy relative to the former hydration step at most hydration degrees, suggesting that hydration overall tends to stabilize the clusters. For SUA-free clusters, the fifth hydration of SA·AM and SA·DMA·AM and the sixth hydration of SA·DMA exhibit positive or nearly zero one step hydration free energies. These endergonic steps are ascribed to the SA·base clusters already being saturated by water molecules in the former hydration steps (Henschel et al., 2013). For SUA-containing clusters, the addition of one more water molecule at a hydration step likely leads to a great rise in free energy, resulting in a relatively large positive value of the one step hydration energy. For example, the one step free energy is 2.99 kcal mol$^{-1}$ for the fifth hydration, 3.24 kcal mol$^{-1}$ for the fourth hydration, and 1.32 kcal mol$^{-1}$ for the third hydration of the SA·DMA·AM·SUA cluster. The large increases in free energies for SA·AM·SUA and



SA·DMA·SUA clusters are explained by structural rearrangement. The positive one-step
hydration energy for the third hydration of SA·DMA·AM·SUA is likely because the hydrate in
the former step (i.e. the dihydrate) is extraordinarily stable.
The relative changes in the free energy due to addition of SUA to the SA·base clusters are
depicted along with hydration degree (Figure 5b). All free energy changes shown in Figure 5b are
negative, confirming that SUA stabilizes the SA·base clusters. For all hydration levels except the
fourth one, the free energy changes for the SA·DMA cluster by SUA addition are more negative
than that for the SA·AM cluster, suggesting that SUA more efficiently stabilizes the hydrated
SA·DMA clusters than the SA·AM cluster. The largest change in the free energy (-7.15 kcal mol$^{-1}$
) between SA·AM·SUA and SA·AM appears at the fourth hydration step, which is attributable to
the structure change due to SUA addition, i.e., an additional hydrogen bond is formed on the AM
molecule in the fourth hydrate of SA·AM·SUA, while such a hydrogen bond is absent in the
SA·AM cluster until the fifth hydration (Figure 1). The largest negative free energy change (-9.86
kcal mol$^{-1}$) in SA·DMA is under the unhydrated condition. The strong hydrogen bonds between
DMA and the two acids formed in the unhydrated SA·DMA·SUA cluster undergo cleavage due
to water uptake, leading to a smaller free energy difference between the SA·DMA·SUA and
SA·DMA clusters with hydration. In addition, the stabilization effect of hydration on the SA·base
clusters is weakened by addition of SUA, particularly for the SA·DMA and SA·DMA·AM clusters,
with much smaller hydration energies for SA·DMA·SUA and SA·DMA·AM·SUA than the
corresponding clusters without SUA (Fig. 5a). The energetic perturbations by SUA addition are
affected by the hydration degree and the base types, implying synergetic interactions among the
different components in the multi-component clusters.



### 3.3 HYDRATION PROFILES


The equilibrium hydrate distributions were calculated for the SA·base clusters with and
without the presence of SUA. Figure 6 displays the relative hydrate distributions under three
typical atmospheric RH values, i.e., 20%, 50% and 80%. The SA·base cluster shows a tendency
to be more extensively hydrated with increasing RH, although the different clusters exhibit
variable characteristics in the hydrate distribution.
Our results for the SA·AM hydrate distribution are consistent with previous studies
(Loukonen et al., 2010; Henschel et al., 2014), showing that the hydrate distribution of SA·AM is
sensitive to the humidity condition (Figure 6a). The completely dry SA·AM cluster dominates the
hydrate distribution under low RH (<40%), while the trihydrate is most prevalent as RH exceeds
40% because of the strong stability of the trihydrate relative to the monohydrate and dihydrate. In
accordance with the evenly spaced hydration energies, the distribution for hydrated SA·DMA
evenly disperse over the unhydrated cluster to dihydrate (Figure 6b). The unhydrated cluster,
monohydrate, and dihydrate together account for 85% of the total population at all RH levels, and
the peak of the hydrate distribution for SA·DMA shows a continuous shift from the unhydrated
cluster to dihydrate as RH increases. The SA·DMA·AM cluster tends to be dehydrated, as reflected
by the fact that the relative population of dry SA·DMA·AM clusters exceeds 50% even under
highly humid conditions (Figure 6c). This suggests that addition of either DMA or AM
considerable lowers the hydrophilicity of SA·AM or SA·DMA. The dehydration trend of the
SA·DMA·AM cluster is similar to previous investigations,[31] in which the base-containing clusters
with SA were found to be less hydrophilic than the SA clusters.
Addition of SUA alters the hydrate distribution of the SA·base clusters. For example, the
hydrate distribution for SA·AM·SUA clusters is slightly broader than that of SA·AM (Figure 6d),



with a considerable population of the fourth hydrate for SA·AM·SUA at high RH. Clearly,
addition of SUA promotes hydration of SA·AM. The broad hydrate distribution is consistent with
the more negative hydration energy at the fourth hydration step for SA·AM·SUA clusters relative
to SA·AM. However, the peaks of the distribution for SA·AM·SUA shift with a similar pattern as
SA·AM with varying RH, i.e., bypassing the monohydrate and dihydrate as the most populated
cluster. The hydrate distribution for SA·DMA·SUA shows distinct characteristics for SA·DMA.
In the presence of SUA, over 80% of the clusters exist in a dry state regardless of the humidity
condition (Figure 6e), indicating that hydration of SA·DMA·SUA is less favorable than that of
SA·DMA. The SA·DMA·SUA hydrate distribution peak at the unhydrated cluster is explained
energetically, since addition of SUA greatly reduces the free energy of the dry clusters and the
changes in hydration free energy are relatively small at all hydration levels. The
SA·DMA·AM·SUA clusters are mostly likely dehydrated or hydrated with two water molecules
depending upon the moisture condition (Figure 6f), as the distribution peak shifts between the
unhydrated cluster (RH < 70%) and the dihydrate (RH >70%). The monohydrate does not exhibit
a maximum of the distribution at any RH level. Clearly, SA·DMA·AM·SUA is more favorably
hydrated compared to SA·DMA·AM.

The hydration profiles as functions of RH for the clusters with SA·base are shown in

(Figure 7). Theoretically, the maximal hydration degrees for SA·AM, SA·DMA, and
SA·DMA·AM clusters on average are 2.7, 1.7, and 0.9 of water molecules, respectively, as RH
approaches 100%. The calculations for the hydration of SA·AM and SA·DMA in this study
slightly overestimate the hydration degree, compared to those by Henschel et al. (2014) The
hydrations for SA·AM, SA·AM·SUA, SA·DMA tend to level off at high RH, but such a trend is
not evident for the other three clusters. With SUA, the hydrophilicity of SA·AM and



SA·DMA·AM systems is considerably enhanced, implying that more water molecules can be
taken up by SUA-containing clusters. In contrast, the number of water molecules that can be bound
to SA·DMA clusters are greatly reduced if SUA is added to the system, in accordance with that
the most populated cluster of SA·DMA·SUA is unhydrated under different moisture conditions
(Figure 6e).
**3.4    ATMOSPHERIC IMPLICATIONS**
The formation of SA·base·SUA with addition of one SUA molecule to the SA·base clusters
is energetically favorable (Figure 5b). SUA is more effective than SA to stabilize the SA·base
clusters. In addition to the formation energy, the relative concentration of the precursor species
involved in clustering also governs the cluster distribution in the atmosphere. To evaluate the
importance of SUA in small cluster formation, the concentration ratios of the SA·base clusters
with uptake of SUA to uptake of an additional SA molecule were determined under
atmospherically-relevant concentrations of SUA and SA, along with unhydrated and hydrated SA
clusters. The calculations of the ratios $[SA·X·SUA]/[(SA)_2·X]$ (X denotes AM, DMA, water
molecule, or none) are based on the thermochemical data in Table S4, as presented in Table 5.
With a high level of SUA, the concentration of SA·AM·SUA cluster is comparable to that
of $(SA)_2·AM$, suggesting that the formation of SA·AM·SUA clusters competes with the formation
of $(SA)_2·AM$ in the atmosphere. $(SA)_2·AM$ dominates the cluster distribution only if SA and SUA
concentrations are at similar levels. The ratio of SA·AM·SUA to $(SA)_2·AM$ is lowered to 1:1000,
when SA concentration is comparable to SUA. SUA-containing clusters are prevalent in the
atmosphere for SA·DMA, since the ratio of $[SA·DMA·SUA]/[(SA)_2·DMA]$ reaches as high as
3000:1 with highly abundant SUA.



The prevalence of SUA-containing clusters is more significant for the unhydrated SA

clusters and the hydrated SA clusters with one water molecule, with the ratios of $[SA \cdot SUA]/[(SA)_2]$

and $[SA \cdot W \cdot SUA]/[(SA)_2 \cdot W]$ of higher than $10^7$:1 and $10^6$:1, respectively. Even when the SUA

concentration is relatively low and comparable to SA, the $[SA \cdot SUA]/[(SA)_2]$ and

$[SA \cdot W \cdot SUA]/[(SA)_2 \cdot W]$ ratios are both higher than 500:1. Sulfuric acid dimer has been

recognized as an important precursor for NPF (Zhang et al., 2012), but our study shows that, as

one of most prevalent dicarboxylic acids in atmosphere, SUA inhibits the formation or further

growth of sulfuric acid dimer because of its strong interaction with SA to stabilize the unhydrated

or hydrated SA clusters. This inhibiting effect of SUA on the formation or further growth of

sulfuric acid dimer is more efficient than ketodiperoxy acid (Elm et al., 2016).

It should be pointed that steady-state equilibrium for pre-nucleation clusters is rarely

established under atmospheric conditions, because of continuous forward reactions by adding

monomers to form larger clusters during NPF. Hence, the ability whether a cluster grows to form

a nano-sized particle is dependent on the competition between the forward reaction by adding a

monomer and the backward reaction by losing a monomer (evaporation) for each intermediate step.

While the evaporation rate relies on the thermodynamic stability of the cluster, the forward rate

constant is kinetically controlled, dependent on the activation and kinetical energies for the

colliding cluster and monomer. For neutral clusters, electrostatic dipole-dipole interaction likely

plays a key role in reducing the activation barrier. The presence of organic acids typically increases

the dipole moment of clusters (Zhao et al., 2009). We calculated an additional subset of clusters

by adding another sulfuric acid molecule to hydrated $SA \cdot DMA \cdot SUA$ clusters, i.e.,

$(SA)_2 \cdot DMA \cdot SUA \cdot (W)_x$. The stable clusters with two SA molecules are depicted in Figure 8. Table

2 also indicates that the Gibbs energies of $(SA)_2 \cdot DMA \cdot SUA \cdot (W)_x$ are negative relative to





SA·DMA·SUA·(W)$_x$, except for the hydrated form with six H$_2$O molecules. These results reveal
that the clusters containing both the base and organic acid are capable of further binding with acid
molecules to promote their growth.
**4.    CONCLUSIONS**

We have investigated the molecular interactions between SUA and SA·base clusters in the

presence of hydration, including AM and DMA. The stable cluster structures were sampled by the
BPMC approach and further geometric optimization and frequency calculation using quantum
chemical calculations at the PW91PW91/6-311++G(2d, 2p) level. The characteristics of the
structures, thermochemistry, and topology of the clusters were analyzed, focusing on the
differences with and without SUA. In addition, the relative hydrate population and the average
hydration numbers for each SA·base cluster were calculated, and the influence of SUA on the
cluster hydration and the competition between SUA and SA to stabilize the SA·base clusters were
assessed.

Addition of SUA to the SA·base cluster systems is energetically favorable at all hydration

levels, suggesting that SUA stabilizes the SA·base clusters and their hydrates. In addition, the
addition of SUA promotes proton transfers in the SA·base clusters. The proton transfer by SUA
addition is confirmed by the formation of new covalent bonds, showing a relocation of the high
LOL value from the SA side to the AM side and a shift from positive to negative for the Laplacian
of electron density. The presence of SUA in SA·AM and SA·DMA clusters generally strengthens
the pre-existing covalent bonds in SA·base·SUA·(W)$_n$ clusters at the various hydration levels,
since the LBO values of the covalent bonds in SUA-containing clusters are higher than those in
the clusters without SUA. The distribution of hydrate population for SA·AM·SUA is broader than
that for SA·AM. Also, the distribution for SA·DMA·AM·SUA hydrates peaks at the two-water





molecule level under elevated RH, but the peak of the distribution for SA·DMA·AM always

corresponds to the unhydrated cluster. The shift in the hydrate distributions to a higher hydration

level in SUA-containing clusters relative to the cluster without SUA suggests that the addition of

SUA enhances the hydrophilicity of SA·AM and SA·DMA·AM. However, the presence of SUA

tends to dehydrate the SA·DMA clusters, since the most prevalent cluster for SA·DMA·SUA is in

a dry state. At equilibrium and considering the typical abundances of SUA and SA in the

atmosphere, the formation of SUA·SA·base (AM or DMA) is comparable to that of $(SA)_2$·base.

Hence, the uptake of SUA competes with the uptake of another SA to stabilize the SA·base clusters,

and the presence of SUA hinders the formation and further growth of SA dimer clusters. The

hydrated SA·DMA·SUA cluster is capable of binding with additional acid molecules, which not

only stabilizes the cluster but also promotes its further growth.

Our results indicate that the multi-component molecular interaction involving organic acids,

sulfuric acid, and base species promotes NPF in the atmosphere, particularly under polluted

environments because of the co-existence of elevated concentrations of these nucleation precursor

species. Future studies are necessary to assess the kinetics (forward and reverse rates) and potential

energy surface of cluster growth, in order to develop parameterization of NPF for atmospheric

models.

**SUPPLEMENTARY MATERIAL**

Supplementary material contains additional relief maps for the sulfuric acid-base clusters

and lists of topological properties for the most stable conformers of each cluster categories.

**ACKNOWLEDGMENTS**

This work was supported by National Natural Science Foundation of China (41675122,

41425015, U1401245, and 41373102), Science and Technology Program of Guangzhou City



(201707010188), Team Project from the Natural Science Foundation of Guangdong Province,
China (S2012030006604), Special Program for Applied Research on Super Computation of the
NSFC-Guangdong Joint Fund (the second phase), National Super-computing Centre in
Guangzhou (NSCC-GZ), and Robert A. Welch Foundation (A-1417). The research was partially
conducted with the advanced computing resources provided by Texas A&M High Performance
Research Computing. The authors acknowledged the Laboratory for Molecular Simulations at
Texas A&M University.



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

A Density Functional Theory Study of the Thermochemistry of Pre-Nucleation Clusters,

Entropy, 13, 554–569, https://doi.org/10.3390/e13020554, 2011.



Ortega, I. K., Kupiainen, O., Kurtén, T., Olenius, T., Wilkman, O., McGrath, M. J., Loukonen, V.,
and Vehkamäki, H.: From quantum chemical formation free energies to evaporation rates,
Atmos. Chem. Phys., 12, 225-235, https://doi.org/10.5194/acp-12-225-2012, 2012
Riccobono, F., Schobesberger, S., Scott, C. E., Dommen, J., Ortega, I. K., Rondo, L., Almeida, J.,
Amorim, A., Bianchi, F., Breitenlechner, M., David, A., Downard, A., Dunne, E. M., Duplissy,
J., Ehrhart, S., Flagan, R. C., Franchin, A., Hansel, A., Junni- nen, H., Kajos, M., Keskinen,
H., Kupc, A., Kürten, A., Kvashin, A. N., Laaksonen, A., Lehtipalo, K., Makhmutov, V.,
Mathot, S., Nieminen, T., Onnela, A., Petäjä, T., Praplan, A. P., Santos, F. D., Schallhart, S.,
Seinfeld, J. H., Sipilä, M., Spracklen, D. V., Stozhkov, Y., Stratmann, F., Tomé, A.,
Tsagkogeorgas, G., Vaattovaara, P., Viisanen, Y., Vrtala, A., Wagner, P. E., Weingart- ner,
E., Wex, H., Wimmer, D., Carslaw, K. S., Curtius, J., Donahue, N. M., Kirkby, J., Kulmala,
M., Worsnop, D. R., and Baltensperger, U.: Oxidation products of biogenic emissions
contribute to nucleation of atmospheric particles, Science 344, 717-721,
https://doi.org/10.1126/science.1243527, 2014.
Qiu, C., and Zhang, R.: Multiphase chemistry of atmospheric amines, Phys. Chem. Chem. Phys.,
15, 5738-5752, https://doi.org/10.1039/C3CP43446J, 2013.
Tröstl, J., Chuang, W. K., Gordon, H., Heinritzi, M., Yan, C., Molteni, U., Ahlm, L., Frege, C.,
Bianchi, F., Wagner, R., Si- mon, M., Lehtipalo, K., Williamson, C., Craven, J. S., Du- plissy,
J., Adamov, A., Almeida, J., Bernhammer, A.-K., Bre- itenlechner, M., Brilke, S., Dias, A.,
Ehrhart, S., Flagan, R. C., Franchin, A., Fuchs, C., Guida, R., Gysel, M., Hansel, A., Hoyle,
C. R., Jokinen, T., Junninen, H., Kangasluoma, J., Kesk- inen, H., Kim, J., Krapf, M., Kürten,
A., Laaksonen, A., Lawler, M., Leiminger, M., Mathot, S., Möhler, O., Nieminen, T., On-
nela, A., Petäjä, T., Piel, F. M., Miettinen, P., Rissanen, M. P., Rondo, L., Sarnela, N.,



Schobesberger, S., Sengupta, K., Sip- ilä, M., Smith, J. N., Steiner, G., Tomè, A., Virtanen,
A., Wag- ner, A. C., Weingartner, E., Wimmer, D., Winkler, P. M., Ye, P., Carslaw, K. S.,
Curtius, J., Dommen, J., Kirkby, J., Kulmala, M., Riipinen, I., Worsnop, D. R., Donahue, N.
M., and Bal- tensperger, U.: The role of low-volatility organic compounds in initial particle
growth in the atmosphere, Nature, 533, 527-531, https://doi.org/10.1038/nature18271, 2016.
Tsona, N. T., Henschel, H., Bork, N., Loukonen, V., and Vehkamäki, H.: Structures, Hydration,
and Electrical Mobilities of Bisulfate Ion–Sulfuric Acid–Ammonia/Dimethylamine Clusters:
A    Computational    Study,    J.    Phys.    Chem.    A,    119,    9670–9679,
https://doi.org/10.1021/acs.jpca.5b03030, 2015.
Wang, L., Khalizov, A.F., Zheng, J., Xu, W., Lal, V., Ma, Y., and Zhang, R.: Atmospheric
nanoparticles formed from heterogeneous reactions of organics, Nature Geosci., 3, 238-242,
https://doi.org/10.1038/ngeo778, 2010.
Wang, J., Krejci, R., Giangrande, S., Kuang, C., Barbosa, H. M. J., Brito, J., Carbone, S., Chi, X.,
Comstock, J., Ditas, F., Lavric, J., Manninen, H. E., Mei, F., Moran-Zuloaga, D., Pöhlker, C.,
Pöh- lker, M. L., Saturno, J., Schmid, B., Souza, R. A. F., Springston, S. R., Tomlinson, J. M.,
Toto, T., Walter, D., Wimmer, D., Smith, J. N., Kulmala, M., Machado, L. A. T., Artaxo, P.,
Andreae, M. O., Petäjä, T., and Martin, S. T.: Amazon boundary layer aerosol concentration
sustained    by    vertical    transport    during    rainfall,    Na- ture,    539,    416–419,
https://doi.org/10.1038/nature19819, 2016.
Wang, C.-Y., Jiang, S., Liu, Y.-R., Wen, H., Wang, Z.-Q., Han, Y.-J., Huang, T., Huang, W.:
Synergistic Effect of Ammonia and Methylamine on Nucleation in the Earth's Atmosphere.
A    Theoretical    Study,    J.    Phys.    Chem.    A,    122,    3470−3479,    https://doi.org/
10.1021/acs.jpca.8b0068, 2018.




Weber, K. H., Liu, Q., and Tao, F.-M.: Theoretical study on stable small clusters of oxalic acid
with ammonia and water, J. Phys. Chem. A, 118, 1451-1468,
https://doi.org/10.1021/jp4128226, 2014.
Wexler, A.: Vapor pressure formulation for water in range 0 to 100 C. A revision, J. Res. Nat. Bur.
Stand. 80A, 775-785, 1976.
Xu, Y., Nadykto, A. B., Yu, F., Jiang, L., and Wang, W.: Formation and properties of hydrogen-
bonded complexes of common organic oxalic acid with atmospheric nucleation precursors, J.
Mol. Struct.: THEOCHEM, 951, 28-33, https://doi.org/10.1016/j.theochem.2010.04.004,
2010a.
Xu, Y., Nadykto, A. B., Yu, F., Herb, J., and Wang, W.: Interaction between common organic
acids and trace nucleation species in the Earth's atmosphere, J. Phys. Chem. A, 114, 387-96,
https://doi.org/10.1021/jp9068575, 2010b.
Xu, W., and Zhang, R.: Theoretical investigation of interaction of dicarboxylic acids with common
aerosol nucleation precursors, J. Phys. Chem., 116, 4539-4550, https://doi.org/
10.1021/jp301964u, 2012.
Xu, W., and Zhang, R.: A theoretical study of hydrated molecular clusters of amines and
dicarboxylic acids, J. Chem. Phys., 139, 064312, https://doi.org/10.1063/1.4817497, 2013.
Xu, W., Gomez-Hernandez, M., Guo, S., Secrest, J., Marrero-Ortiz, W., Zhang, A. L., and Zhang,
R.: Acid-catalyzed reactions of epoxides for atmospheric nanoparticle growth, J. Am. Chem.
Soc., 136, 15477−15480, https://doi.org/10.1021/ja508989a, 2014.
Yu, H., McGraw, R., and Lee, S.-H.: Effects of amines on formation of sub-3 nm particles and
their subsequent growth, Geophys. Res. Lett., 39, L02807,
https://doi.org/10.1029/2011GL050099, 2012.



Yue, D. L., Hu, M., Zhang, R. Y., Wang, Z. B., Zheng, J., Wu, Z. J., Wiedensohler, A., He, L. Y.,
Huang, X. F., and Zhu, T.: The roles of sulfuric acid in new particle formation and growth in
the mega-city of Beijing, Atmos. Chem. Phys. 10, 4953–4960, https://doi.org/ 10.5194/acp-

655 10-4953-2010, 2010.

Yue, D. L., Hu, M., Zhang, R. Y., Wu, Z. J., Su, H., Wang, Z. B., and Wiedensohler, A.: Potential
contribution of new particle formation to cloud condensation nuclei in Beijing, Atmos.
Environ., 45, 6070-6077, https://doi.org/ 10.1016/j.atmosenv.2011.07.037, 2011.
Yao, L., Garmash, O., Bianchi, F., Zheng, J., Yan, C., Kontkanen, J., Junninen, H., Mazon, B. S.,
Ehn, M., Paasonen, P., Sipila, M., Wang, M., Wang, X., Xiao, S., Chen, H., Lu, Y., Zhang,
B., Wang, D., Fu, Q., Geng, F., Li, L., Wang, H., Qiao, L., Yang, X., Chen, J., Kerminen, V.,
Petaja, T., Worsnop, D., Kulmala, M., Wang, L.: Atmospheric new particle formation from
sulfuric acid and amines in a Chinese megacity, Science, 361, 278-281,
https://doi.org/10.1126/science.aao4839, 2018.
Zhang, H., Kupiainen-Määttä, O., Zhang, X., Molinero, V., Zhang, Y., and Li, Z., The
enhancement mechanism of glycolic acid on the formation of atmospheric sulfuric acid–
ammonia molecular clusters, J. Chem. Phys., 146, 184308, https://doi.org/10.1063/1.4982929,

2017.

Zhang, R.: Getting to the critical nucleus of aerosol formation, Science, 328, 1366-1367,
https://doi.org/10.1126/science.1189732, 2010.
Zhang, R., Suh, I., Zhao, J., Zhang, D., Fortner, E. C., Tie, X., Molina, L. T., and Molina, M. J.:
Atmospheric new particle formation enhanced by organic acids, Science, 304, 1487-1490,
https://doi.org/10.1126/science.1095139, 2004.




Zhang, R., Wang, L., Khalizov, A. F., Zhao, J., Zheng, J., McGraw, R. L., and Molina, L. T.:

Formation of nanoparticles of blue haze enhanced by anthropogenic pollution, Proc. Natl.

Acad. Sci. USA, 106, 17650-17654, https://doi.org/10.1073/pnas.0910125106, 2009.

Zhang, R., Khalizov, A.F., Wang, L., Hu, M., Xu, W.: Nucleation and growth of nanoparticles in

the atmosphere, Chem. Rev., 112, 1957-2011, https://doi.org/10.1021/cr2001756, 2012.

Zhang, R., Wang, G., Guo, S., Zamora, M. L., Ying, Q., Lin, Y., Wang, W., Hu, M., and Wang Y.:

Formation of urban fine particulate matter, Chem. Rev., 115, 3803-3855,

https://doi.org/10.1021/acs.chemrev.5b00067, 2015.

Zhao, J., Khalizov, A., Zhang, R., and McGraw, R.: Hydrogen bonding interaction of molecular

complexes and clusters of aerosol nucleation precursors, J. Phys. Chem. A 113, 680–689,

https://doi.org/10.1021/jp806693r, 2009.

Zhu, Y. P., Liu, Y. R., Huang, T., Jiang, S., Xu, K. M., Wen, H., Zhang, W. J., and Huang, W.:

Theoretical study of the hydration of atmospheric nucleation precursors with acetic acidJ,

Phys. Chem. A, 118, 7959-7974, https://doi.org/10.1021/jp506226z, 2014.




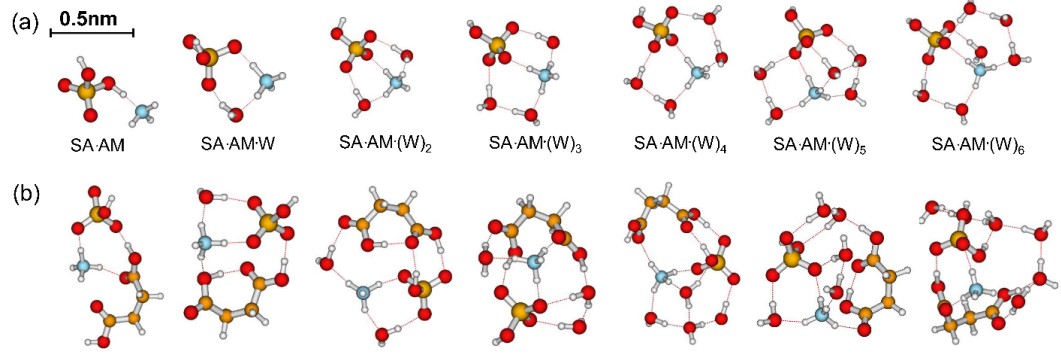


FIG. 1. Most stable configurations of the hydrated SA•AM clusters and the clusters with one
SUA addition. The hydration is with 0-6 water molecules. The sulfur (carbon) atoms are depicted
as large (small) yellow balls, oxygen atoms in red, nitrogen atoms in blue, and hydrogen atoms
in white. The dash line denotes the hydrogen bond.



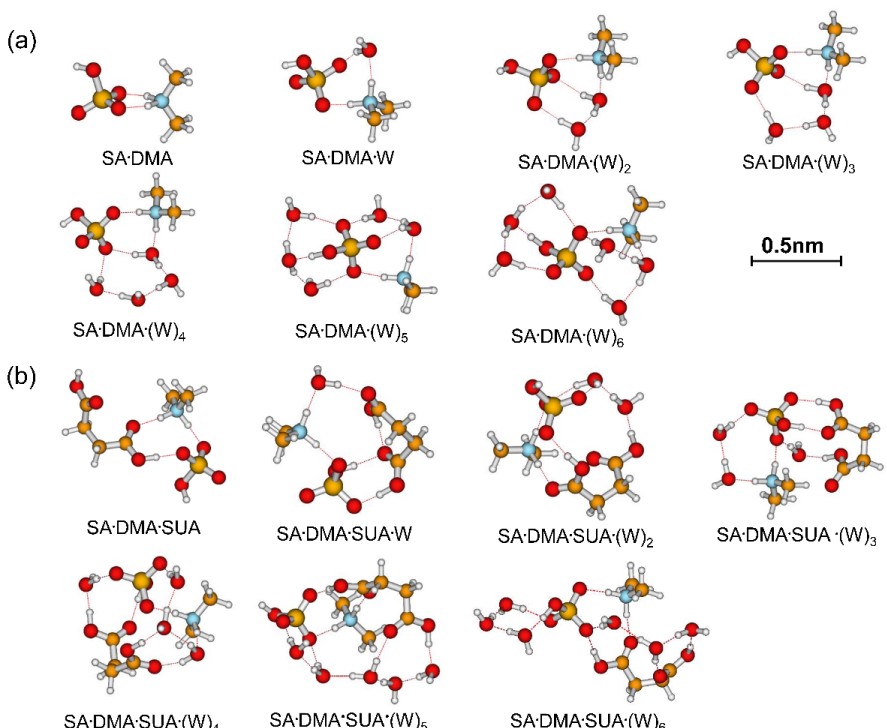

FIG. 2. Most stable configurations of the hydrated SA•DMA clusters and the clusters with one

SUA addition. The hydration is with 0-6 water molecules.



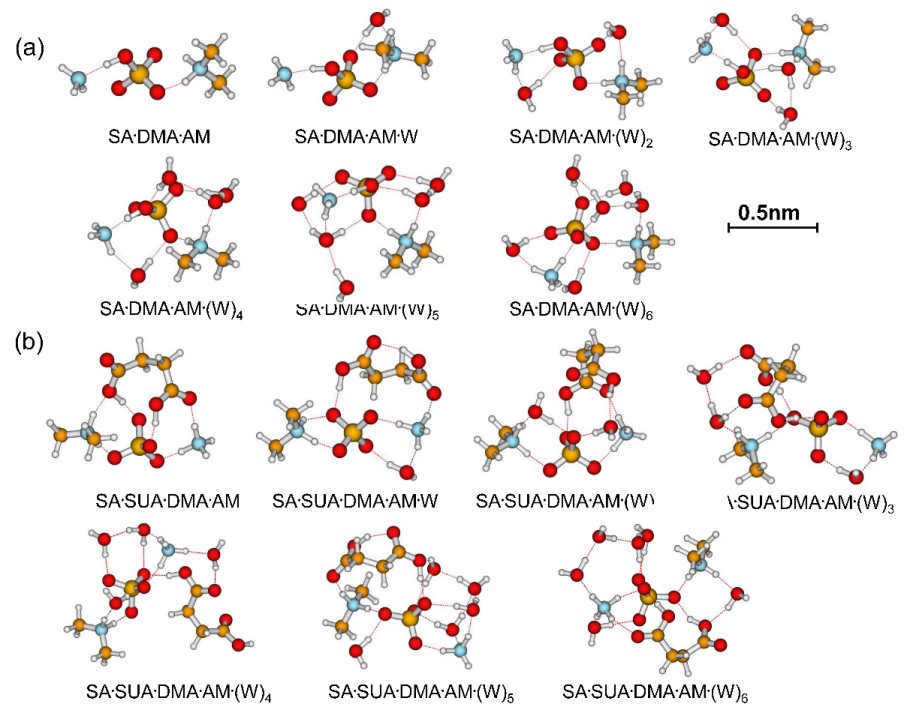


FIG. 3. Most stable configurations of the hydrated SA•DMA•AM clusters and the clusters with
one SUA addition. The hydration is with 0-6 water molecules.



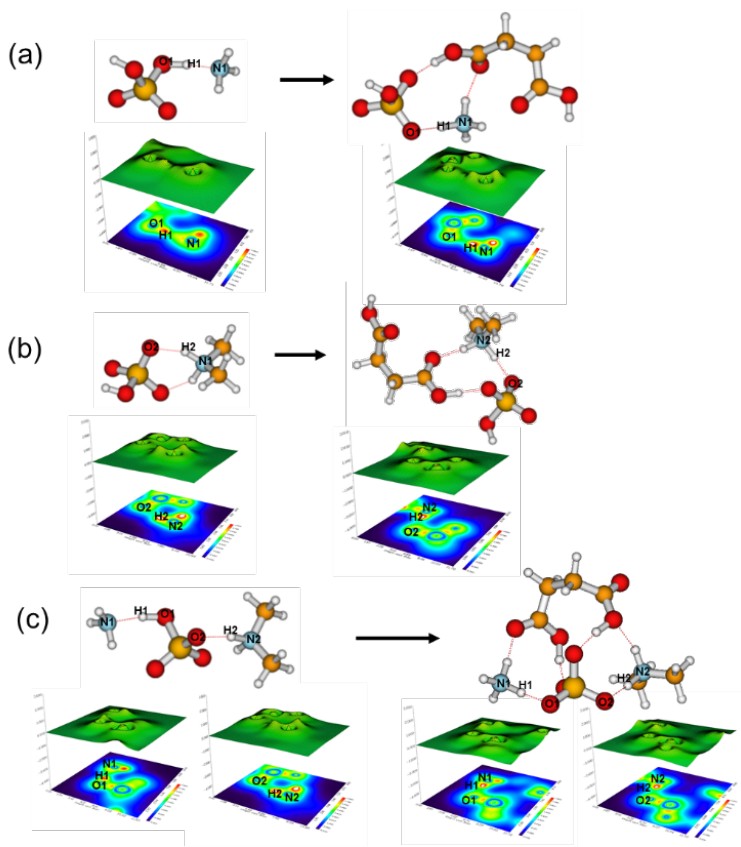


FIG. 4. Relief maps with the projection of localized orbital locator for clusters of (a) SA·AM and

SA·AM·SUA, (b) SA·DMA, SA·DMA·SUA, and (c) SA·DMA·AM and SA·DMA·AM·SUA.

Hydrogen bonds are shown as dashed lines. A large LOL value reflects that electrons are greatly

localized, indicating the existence of a covalent bond.





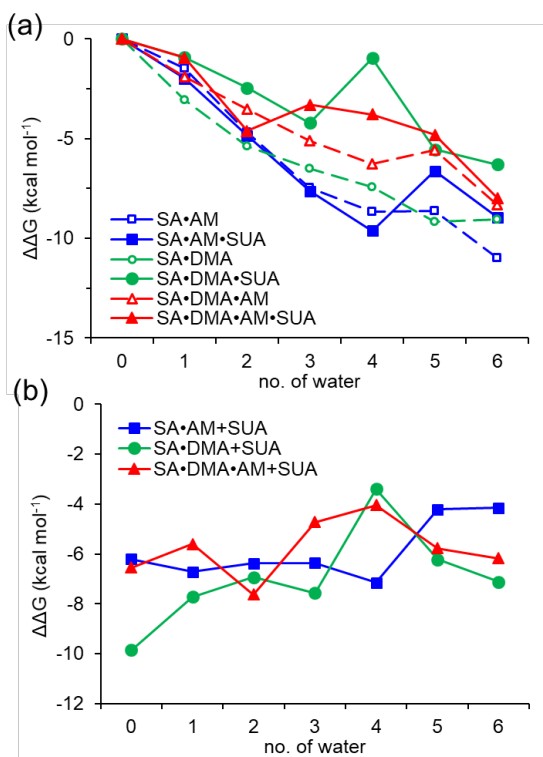


FIG. 5. Hydration free energies (a) and the relative Gibbs free energy changes due to addition of
one SUA molecule to SA•base clusters (b) at $T$=298.15 K and $p$=1 atm. The free energy is
calculated at the PW91PW91/6-311++G(2d, 2p) level.




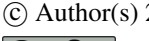

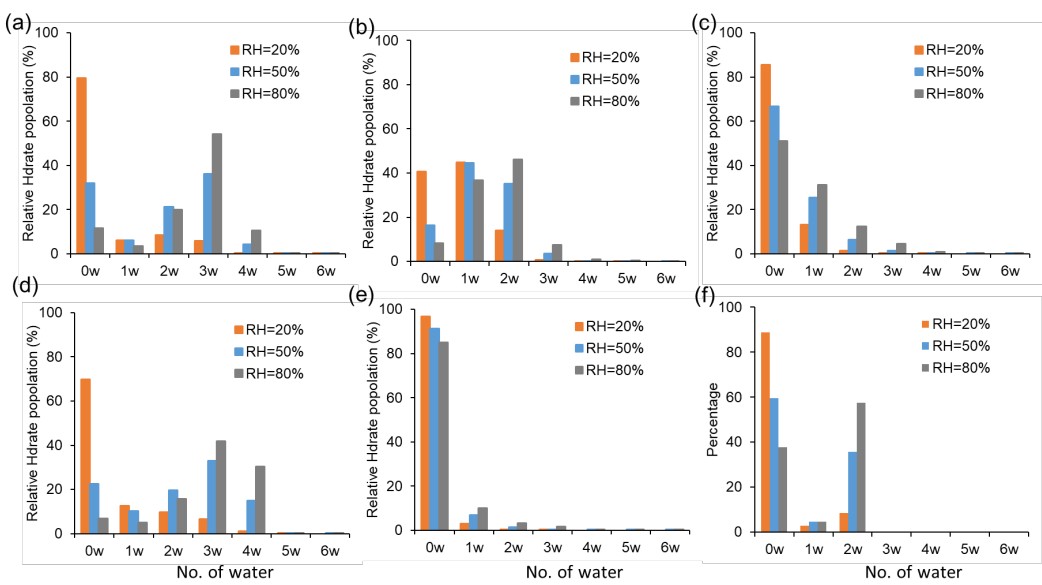


FIG. 6. Hydrate distributions of clusters under different RH levels (20%, 50% and 80%). (a), (b),
and (c) are clusters for SA•AM, SA•DMA, and SA•DMA•AM, respectively. (d), (e) and (f) are
clusters with one SUA addition on the basis of (a), (b) and (c) clusters. In all RH cases, T=298 K.





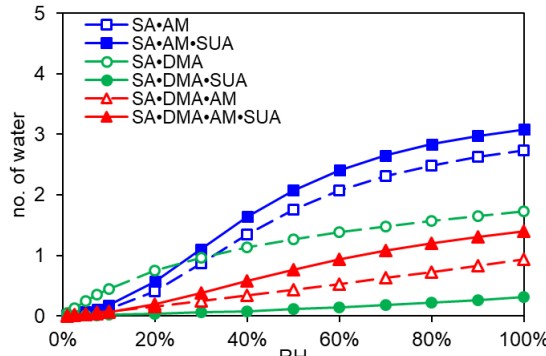


FIG. 7. Average hydration numbers per cluster for various SA·base clusters at 298.15 K.



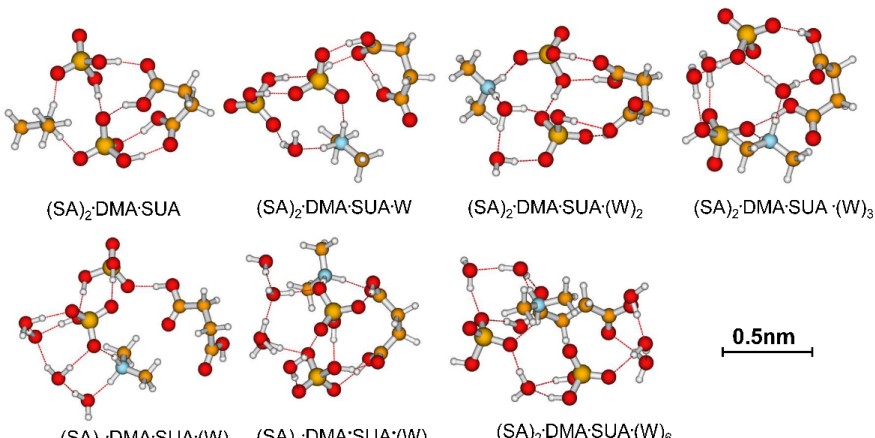


FIG. 8. Most stable configurations of the hydrated (SA)$_2$•DMA•SUA clusters. The hydration is
with 0-6 water molecules (W).




Table 1. Theoretical and experimental values of the free energy change for several basic
reactions in kcal mol$^{-1}$.

| Reactions | This study | | refs |
|---|---|---|---|
| | PW91PW91/6-311++G(2df,2pd) | M06-2X/6-311++G(3df,3pd) | |
| SA+AM →SA•AM | -7.65 | -8.00 | -8.5[a,*], -7.77[b], -6.64[c], -7.84[d] |
| SA+DMA→SA•DMA | -11.13 | -11.24 | 13.66[c], 11.38[e] |
| SA•AM+W→SA•AM•W | -1.48 | -0.07 | -1.41[b], -1.67[f] |
| SA•DMA+W→SA•DMA•W | -3.06 | -3.63 | -3.67[e], -2.89[f] |
| SA•AM+SUA→SA•AM•SUA | -6.20 | -7.29 | - |
| SA•DMA+SUA →SA•DMA•SUA | -9.86 | -11.46 | - |

[a] From Hanson and Eisele (2002)
[b] From Nadykto and Yu (2007)
[c] From Kurtén et al. (2008)
[d] From Elm et al. (2012)
[e] From Nadykto et al. (2011)
[f] From Henschel et al. (2014)
[*] corresponds to experimental results.



Table 2. Calculated binding energy $\Delta E(ZPE)$, enthalpy $\Delta H$, and Gibbs free energy $\Delta G$ (at
$T$=298.15 K and $p$=1 atm) at the PW91PW91/6-311++G(2d, 2p) level of theory for the hydrated
clusters. Energies are in kcal mol$^{-1}$.

| Reactions | $\Delta E(ZPE)$ | $\Delta H$ | $\Delta G$ |
|---|---|---|---|
| SA+AM → SA•AM | -15.95 | -16.51 | -7.65 |
| SA+AM+W → SA•AM•W | -26.40 | -28.02 | -9.13 |
| SA+AM+2W → SA•AM•(W)$_2$ | -39.14 | -41.58 | -12.33 |
| SA+AM+3W → SA•AM•(W)$_3$ | -49.96 | -52.98 | -15.11 |
| SA+AM+4W → SA•AM•(W)$_4$ | -59.63 | -63.30 | -16.32 |
| SA+AM+5W → SA•AM•(W)$_5$ | -69.20 | -73.58 | -16.26 |
| SA+AM+6W → SA•AM•(W)$_6$ | -78.71 | -83.37 | -18.64 |
| | | | |
| SA+DMA → SA•DMA | -21.16 | -21.11 | -11.13 |
| SA+DMA+W →SA• DMA•W | -33.46 | -34.13 | -14.19 |
| SA+DMA+2W → SA•DMA•(W)$_2$ | -45.02 | -46.60 | -16.51 |
| SA+DMA+3W → SA•DMA•(W)$_3$ | -54.37 | -56.57 | -17.62 |
| SA+DMA+4W → SA•DMA•(W)$_4$ | -62.47 | -65.25 | -18.56 |
| SA+DMA+5W → SA•DMA•(W)$_5$ | -75.68 | -79.92 | -20.31 |
| SA+DMA+6W → SA•DMA•(W)$_6$ | -84.43 | -89.24 | -20.16 |
| | | | |
| SA+SUA+AM → SA•SUA•AM | -34.19 | -34.69 | -13.85 |
| SA+SUA+AM+W → SA•SUA•AM•W | -45.65 | -47.18 | -15.85 |
| SA+SUA+AM+2W → SA•SUA•AM•(W)$_2$ | -58.95 | -61.44 | -18.70 |
| SA+SUA+AM+3W → SA•SUA•AM•(W)$_3$ | -70.41 | -73.52 | -21.47 |
| SA+SUA+AM+4W → SA•SUA•AM•(W)$_4$ | -80.92 | -84.95 | -23.47 |
| SA+SUA+AM+5W → SA•SUA•AM•(W)$_5$ | -86.75 | -90.96 | -20.48 |
| SA+SUA+AM+6W → SA•SUA•AM•(W)$_6$ | -98.52 | -104.27 | -22.80 |
| | | | |
| SA+SUA+DMA → SA•SUA•DMA | -42.01 | -41.42 | -20.98 |
| SA+SUA+DMA+W → SA•SUA•DMA•W | -54.80 | -55.47 | -21.92 |
| SA+SUA+DMA+2W → SA•SUA•DMA•(W)$_2$ | -64.86 | -66.03 | -23.45 |
| SA+SUA+DMA+3W → SA•SUA•DMA•(W)$_3$ | -75.90 | -78.21 | -25.19 |
| SA+SUA+DMA+4W → SA•SUA•DMA•(W)$_4$ | -82.83 | -86.21 | -21.95 |
| SA+SUA+DMA+5W → SA•SUA•DMA•(W)$_5$ | -92.80 | -96.19 | -26.54 |
| SA+SUA+DMA+6W → SA•SUA•DMA•(W)$_6$ | -103.04 | -107.49 | -27.29 |
| | | | |
| 2SA+SUA+DMA → (SA)$_2$•SUA•DMA | -62.90 | -63.35 | -26.12 |
| 2SA+SUA+DMA+W → (SA)$_2$•SUA•DMA•W | -69.95 | -70.90 | -25.11 |
| 2SA+SUA+DMA+2W → (SA)$_2$•SUA•DMA•(W)$_2$ | -79.07 | -80.72 | -25.30 |
| 2SA+SUA+DMA+3W → (SA)$_2$•SUA•DMA•(W)$_3$ | -91.67 | -94.06 | -28.71 |
| 2SA+SUA+DMA+4W → (SA)$_2$•SUA•DMA•(W)$_4$ | s-93.90 | -96.57 | -24.36 |
| 2SA+SUA+DMA+5W → (SA)$_2$•SUA•DMA•(W)$_5$ | -115.58 | -120.45 | -31.69 |





| | | | |
|---|---|---|---|
| 2SA+SUA+DMA+6W → (SA)$_2$•SUA•DMA•(W)$_6$ | -108.55 | -112.07 | -22.50 |
| | | | |
| SA+DMA+AM → SA•DMA•AM | -23.83 | -33.01 | -14.15 |
| SA+DMA+AM+W → SA•DMA•AM•W | -44.15 | -45.58 | -16.05 |
| SA+DMA+AM+2W → SA•DMA•AM•(W)$_2$ | -54.34 | -56.54 | -17.69 |
| SA+DMA+AM+3W → SA•DMA•AM•(W)$_3$ | -66.01 | -69.12 | -19.27 |
| SA+DMA+AM+4W → SA•DMA•AM•(W)$_4$ | -75.88 | -79.62 | -20.44 |
| SA+DMA+AM+5W → SA•DMA•AM•(W)$_5$ | -83.63 | -87.99 | -19.74 |
| SA+DMA+AM+6W → SA•DMA•AM•(W)$_6$ | -97.07 | -102.91 | -22.48 |
| | | | |
| SA+SUA+DMA+AM → SA•SUA•DMA•AM | -54.69 | -56.03 | -20.69 |
| SA+SUA+DMA+AM+W → SA•SUA•DMA•AM•W | -62.07 | -63.89 | -21.65 |
| SA+SUA+DMA+AM+2W → SA•SUA•DMA•AM•(W)$_2$ | -77.31 | -80.08 | -25.32 |
| SA+SUA+DMA+AM+3W → SA•SUA•DMA•AM•(W)$_3$ | -83.64 | -87.00 | -24.00 |
| SA+SUA+DMA+AM+4W → SA•SUA•DMA•AM•(W)$_4$ | -92.14 | -95.95 | -24.48 |
| SA+SUA+DMA+AM+5W → SA•SUA•DMA•AM•(W)$_5$ | -104.97 | -110.11 | -25.51 |
| SA+SUA+DMA+AM+6W → SA•SUA•DMA•AM•(W)$_6$ | -115.86 | -121.79 | -28.66 |





Table 3. Number of Proton Transfers within hydrated Clusters (T = 298.15 K).

| Cluster | No. of water | | | | | | |
|---|---|---|---|---|---|---|---|
| | *0* | *1* | *2* | *3* | *4* | *5* | *6* |
| SA[a] | 0 | 0 | 0 | 1 | 1 | 1 | 1 |
| SA•AM | 0 | 1 | 1 | 1 | 1 | 1 | 1 |
| SA•AM•SUA | 1 | 1 | 1 | 1 | 1 | 1 | 1 |
| SA•DMA | 1 | 1 | 1 | 1 | 1 | 1 | 1 |
| SA•DMA•SUA | 1 | 1 | 1 | 1 | 1 | 1 | 1 |
| SA•DMA•AM | 1 | 1 | 1 | 1 | 1 | 1 | 2 |
| SA•DMA•AM•SUA | 2 | 2 | 2 | 2 | 2 | 2 | 2 |

[a] From Xu and Zhang (2013)



Table 4. Laplacian bond order (LBO) of the newly formed covalent bond (nitrogen-hydrogen
bond) between in the clusters (a.u.).

| Clusters | Bonds | No. of water | | | | | | |
|---|---|---|---|---|---|---|---|---|
| | | *0* | *1* | *2* | *3* | *4* | *5* | *6* |
| SA·AM | N1-H1 | - | 0.383 | 0.577 | 0.586 | 0.580 | 0.636 | 0.663 |
| SA·AM·SUA | N1-H1 | 0.464 | 0.575 | 0.586 | 0.621 | 0.609 | 0.663 | 0.607 |
| SA·DMA | N2-H2 | 0.542 | 0.503 | 0.571 | 0.571 | 0.577 | 0.579 | 0.610 |
| SA·DMA·SUA | N2-H2 | 0.551 | 0.548 | 0.598 | 0.613 | 0.583 | 0.613 | 0.581 |
| SA·DMA·AM | N1-H1 | - | - | - | - | - | - | 0.525 |
| | N2-H2 | 0.489 | 0.483 | 0.608 | 0.553 | 0.533 | 0.591 | 0.567 |
| SA·DMA·AM·SUA | N1-H1 | 0.420 | 0.521 | 0.483 | 0.321 | 0.607 | 0.591 | 0.677 |
| | N2-H2 | 0.498 | 0.411 | 0.518 | 0.611 | 0.501 | 0.564 | 0.568 |

Note: N1 is the nitrogen atom on the ammonia (AM) molecule; N2 is the nitrogen atom on the
dimethylamine (DMA) molecule; H1 is the hydrogen atom on one of the hydroxyl functions of sulfuric
acid (SA) molecule and bound to N1; H2 is the hydrogen atom on one of the hydroxyl functions of SA
(SA) molecule and bound to N2.






Table 5. Concentration Ratios between SUA•SA•X and $(SA)_2$•X Clusters, with X = W, AM, and
DMA.

| SUA/SA | X=(None) | X=W | X=AM | X=DMA |
|---|---|---|---|---|
| 1:1 | 3.80E+03 | 5.30E+02 | 4.11E-03 | 3.19E-01 |
| 10:1 | 3.80E+04 | 5.30E+03 | 4.11E-02 | 3.19E+00 |
| 100:1 | 3.80E+05 | 5.30E+04 | 4.11E-01 | 3.19E+01 |
| 1000:1 | 3.80E+06 | 5.30E+05 | 4.11E+00 | 3.19E+02 |
| 10 000:1 | 3.80E+07 | 5.30E+06 | 4.11E+01 | 3.19E+03 |
