# Peer review of "Interaction between Dicarboxylic Acid and Sulfuric Acid-Base Clusters Enhances New Particle Formation"

_Atmospheric Chemistry and Physics, 2018_

## Referee Comment (RC1) · Anonymous Referee #1 · 30 Oct 2018

Yun Lin and co-workers have used computational methods to study clusters containing sulfuric acid (SA), ammonia (AM), dimethylamine (DMA), succinic acid (SUA) and water. The computational methods used are adequate for the purpose (if a bit outdated in terms of the DFT method used to obtain the thermodynamics - the sampling approach on the other hand is state-of-the-art). The purpose of the study is to assess the possible atmospheric significance of succinic acid in promoting sulfuric acid - based new-particle formation (and indeed a positive conclusion is implied even in the title). The study provides important new insights and information onto e.g. the hydration behaviour of SA-AM-DMA-SUA clusters, and especially on how this changes with the presence of SUA. Some of the conclusions concerning the effect (or lack of

effect) of SUA on new-particle formation are somewhat premature, and not always fully supported (indeed, sometimes flat-out contradicted) by the actual data. While the manuscript is publishable, extensive rewriting and reformulation is thus necessary prior to publication in ACP.

Major comments:

1)Overall, the authors attach way too much importance to the sign of the standard Gibbs free energy change, "delta-G". (This is unfortunately very common in similar studies.) The free energies are computed using a 1 atm reference pressure (values computed with other reference pressures would be very different for reactions where the number of molecules changes, as in all the clustering reactions studied here). The concentrations of the studied reactants are (except for water), far far below 1 atm, as the authors themselves note. In other words, the reference pressure is essentially (for these reactions) an arbitrary value that has very little to do with the particular reactions being studied. A negative "delta-G" value for a clustering reaction is a necessary, but very far from sufficient, criterion for the cluster to be said to be "stable". Consider for example a reaction of the type X + Y <=> XY where X is some core cluster, and Y is a molecule present at ppt - levels (i.e. a partial pressure of 1E-12 atm, typical for e.g. SA, DMA or SUA in this study). If the delta-G for this reaction is for example -3 kcal/mol, we can use the law of mass balance to write (here R is the gas constant, T=298K is the temperature, e.g. pXY is the partial pressure of XY, and pref is the reference pressure at which delta-G is computed, in this study 1 atm):

(pXY/pref) / ((pX/pref)(pY/pref)) = exp(-deltaG/RT)

We can solve for the ratio (pXY/pX), and get (pY/pref) exp(-deltaG/RT)

Since pY/pref is 1E-12, and exp(-deltaG/RT) is (for a delta-G of about -3 kcal/mol) about 160, we get a value of about 1.6E-10 for the ratio. In other words less than one in a billion of the molecules of type Y will be bound to XY clusters at equilibrium, despite a negative delta-G value. Clearly, XY is not "stable" in any meaningful sense of the word,

and Y does not "stabilise" X, despite the standard free energy of addition of Y to X being negative. Thus, sentences such as (line 285-286) "All free energy changes shown in Figure 5b are negative, confirming that SUA stabilizes the SAÂůbase clusters" are absolutely false: negative free energy changes do not in any way confirm "stabilisation". (Also, even when a molecule Y is very strongly bound to a cluster X, this does not in and of itself prove "stabilisation" of the cluster as a whole: it just proves that it is difficult for molecule Y to evaporate. Other constituents of the cluster may evaporate either more or less easily due to the presence of Y, and this needs to be separately evaluated - just looking at the Y to X binding is not sufficient to claim "stabilization".) I recommend that the authors search the manuscript for all occurrences of the word "stable" in any form (verb or adjective), and rigorously consider whether or not its use is justified. My guess is that almost none of the claims of "stability" or "stabilisation" are actually really justified (apart from perhaps the discussion on hydration, where the effect of H2O concentration is properly accounted for) - the binding of SUA to the clusters tends to be quite weak, with the equilibrium strongly on the side of the reactants even for the highest SUA concentrations claimed by the authors. (See Elm et al., 2017, also cited in the manuscript, for a discussion on what is actually required for a cluster to be "stable" given trace-gas concentration levels of the constituent monomers: this typically means delta-G values far below -10 kcal/mol - which is not reached for any of the SUA addition steps in the manusript.)

2)Related to the previous point, the most interesting part of the data in terms of evaluating the role of SUA on new-particle formation are the energetics for the formation of the clusters shown in Figure 8, i.e. the clusters with two SA molecules. From table 2, it seems that the SA addition free energies to the (SA)(SUA)(DMA)(W)n clusters vary between +2 and -6 kcal/mol, with the value for the unhydrated cluster being -5.14 kcal/mol. This is significantly HIGHER (more positive) than the free energies of SA addition to the (SA)(DMA) cluster or the (SA)2DMA cluster (without any SUA), which are about -18 and -9 kcal/mol (respectively) according to the coupled-cluster (DLPNO-CCSD(T)) study of Myllys et al https://pubs.acs.org/doi/suppl/10.1021/acs.jpca.5b09762), and even more

negative according to older DFT - or RICC2- based - studies. (In the absence of data at the exact level used by the authors of this study a direct comparison is impossible - the authors might want to compute at least the unhydrated SA(2)DMA and possibly (SA)3DMA clusters just to check.) Thus, the data presented here indicates - in direct contrast to the manuscript title - that the presence of SUA actually HINDERS sulfuric acid - DMA nucleation rather than enhancing it. I strongly urge the authors to reformulate their title in light of this observation (as well as the comments on stability presented above). Also claims such as SUA "promoting subsequent growth" seem unlikely to be true, if the net effect of SUA is to decrease rather than increase the uptake of further SA molecules.

3)The authors present somewhat misleading concentration ranges and ratios for SUA and SA. The SA value of 1E5 molecules per cm3 corresponds to fairly clean conditions - in polluted environments the SA concentration can easily be a couple orders of magnitude higher. The SUA concentration of 1E7 quoted is from Los Angeles, which I presume corresponds to fairly polluted conditions (with SA certainly exceeding 1E5!) - in clean conditions the SUA concentration is very likely much lower. The total organic acid concentration range of 1E8...1E9 quoted is valid, BUT the majority of these will be simple monocarboxylic acids - dicarboxylics such as SUA will only account for a small fraction of the total. Thus, the [SUA]/[SA] range used in the paper (1 to 10 000) is obtained by combining minimum values for SA with maximum values for SUA - in reality, the ratio may well be below one in most places, and I find it hard to believe that values of 10 000 will be found anywhere in the atmosphere. Using the range 1 to 10 000 is fine for answering the question "could SUA possibly play a role in NPF anywhere in the atmosphere, even in the best case", but then this should be stated openly, instead of implying that the range used is representative for most areas of the world. Again, together with the two previous issues, this indicates that the statement in the manuscript title is exaggerated at best, and false at worst.

Minor comments:

-In the BPMC conformational sampling, what force field was used for the SA, AM, DMA and SUA molecules? (For water TIP3P was apparently used.) Presumably some variant of the AMBER force field, but this should be stated.

-In reaction equations (e.g. 4, 5), arrows (going in both directions to indicate a reversible reaction) should be used rather than equals signs

-On line 218, please specify that the proton transfer occurring upon hydration of (SA)(AM)(DMA)(W5) is the SECOND proton transfer, forming the sulfate dianion (overall, it would be good to separate discussion of first and second proton transfers).

-It's a bit unclear what's being plotted in Figure 5a, apparently the free energy of the reaction X + nH2O => X(H2O)n where X is some core cluster? This could be explicitly specified. Or even better, plot the stepwise hydration energies instead, as the actual hydration in the atmosphere likely involves addition (or removal) of single water molecules, not e.g. 5 molecules at the same time. (Since the stepwise energies are then referred to in discussing the hydrate equilibria, it would be better to plot them in the first place.)

---

## Referee Comment (RC2) · Anonymous Referee #3 · 10 Jan 2019

This manuscript presents a theoretical study on the interactions between succinic acid (SUA) and sulfuric acid (SA) – ammonia (AM)/dimethylamine (DMA) clusters in presence of water molecules. The application of the results in atmospheric new particle formation (NPF) is discussed. Overall, this study tackles how a multi-component system, which is more realistic in terms of atmospheric NPF, evolves and provides some of the novel insights into the interaction between organic acids and SA-base clusters. On the other hand, the results and their implication could have been presented in a way that is easier to be followed. The authors are advised to address the following concerns before a recommendation can be made.

1. The atmospheric concentrations of NPF precursors are especially important when one wants to discuss the implication of a theoretical calculation. In fact, the authors failed to find a reliable source for the key species that appear in this study. The concentration of SUA is referred from Kawamura and Kaplan, 1987, which actually presents concentrations of particulate SUA and should not be used in a clustering system. The concentration of SA is set at $10^5$ molecule $cm^{-3}$, which is at least one order of magnitude lower than many measurement values.

2. In many cases, the authors compare delta(G) and then conclude that SUA (and/or other molecules) either promote or hinder the growth of clusters. This is fine when the difference between two delta(G) is large. On the other hand, one probably wants to include the concentrations of gaseous precursors and clusters, do the math, and then obtain something like a branching ratio when the difference is small, instead of simply using "promotion or hindrance". Following this point, I would like to see a clear definition of "promotion or hindrance" in the manuscript. Is it a comparison between the current step of reaction/clustering or a comparison of the further growth of formed cluster from the current step?

3. Given the comprehensive calculation that has been performed, I suppose the authors could suggest a pathway (or multiple possibilities with relative weight for each) on how the complex clusters are formed? e.g., how is (SA)(DMA)(AM)(SUA)(W)6 formed? Will water be added to cluster at the beginning? A summary like this would be welcome, even for a smaller cluster if such a pathway is too complex for a big cluster.

Minor comments, 4. The reference list should be carefully checked. Some of the references are not in an alphabetical order. Also, I suppose what in Line 75 should be "Xu and Zhang (2012)".

---

## Author Comment (AC1) · 7 Mar 2019

see attached

Please also note the supplement to this comment:
https://www.atmos-chem-phys-discuss.net/acp-2018-975/acp-2018-975-AC1-supplement.pdf

---

## Author Response (AR1)

**Manuscript acp-2018-975**

Dear Jingkun,

Please find enclosed our revised manuscript and the responses to the two reviewers.

We thank you for your effort in guiding this review process and greatly appreciate the thoughtful and helpful suggestions from the two reviewers in improving our manuscript. We believe that the revised paper has satisfactorily addressed the concerns from the reviewers. Below please find our point-by-point responses to the reviewer's comments.

Sincerely

Renyi Zhang

**1. Review #1**

**Summary:** Yun Lin and co-workers have used computational methods to study clusters containing sulfuric acid (SA), ammonia (AM), dimethylamine (DMA), succinic acid (SUA) and water. The computational methods used are adequate for the purpose (if a bit outdated in terms of the DFT method used to obtain the thermodynamics - the sampling approach on the other hand is state-of-the-art). The purpose of the study is to assess the possible atmospheric significance of succinic acid in promoting sulfuric acid - based new-particle formation (and indeed a positive conclusion is implied even in the title). The study provides important new insights and information onto e.g. the hydration behaviour of SA-AM-DMA-SUA clusters, and especially on how this changes with the presence of SUA. Some of the conclusions concerning the effect (or lack of C1 ACPD Interactive comment Printer-friendly version Discussion paper effect) of SUA on new-particle formation are somewhat premature, and not always fully supported (indeed, sometimes flat-out contradicted) by the actual data. While the manuscript is publishable, extensive rewriting and reformulation is thus necessary prior to publication in ACP.

**Thanks the reviewer for insightful comments.**

**Major comments:**
**-Comment #1.** Overall, the authors attach way too much importance to the sign of the standard Gibbs free energy change, "delta-G". (This is unfortunately very common in similar studies.) The free energies are computed using a 1 atm reference pressure (values computed with other reference pressures would be very different for reactions where the number of molecules changes, as in all the clustering reactions studied here). The concentrations of the studied reactants are (except for water), far far below 1 atm, as the authors themselves note. In other words, the reference pressure is essentially (for these reactions) an arbitrary value that has very little to do with the particular reactions being studied. A negative "delta-G" value for a clustering reaction is a necessary, but very far from sufficient, criterion for the cluster to be said to be "stable". Consider for example a reaction of the type $X + Y <=> XY$ where X is some core cluster, and Y is a molecule present at ppt - levels (i.e. a partial pressure of 1E-12 atm, typical for e.g. SA, DMA or SUA in this study). If the delta-G for this reaction is for example -3 kcal/mol, we can use the law of mass balance to write (here R is the gas constant, T=298K is the temperature, e.g. pXY is the partial pressure of XY, and pref is the reference pressure at which delta-G is computed, in this study 1 atm):
$$(pXY/pref) / ((pX/pref)(pY/pref)) = exp(-deltaG/RT)$$
We can solve for the ratio (pXY/pX), and get (pY/pref) exp(-deltaG/RT). Since pY/pref is 1E-12, and exp(-deltaG/RT) is (for a delta-G of about -3 kcal/mol) about 160, we get a value of about 1.6E-10 for the ratio. In other words less than one in a billion of the molecules of type Y will be bound to XY clusters at equilibrium, despite a negative delta-G value. Clearly, XY is not "stable" in any meaningful sense of the word, C2 and Y does not "stabilise" X, despite the standard free energy of addition of Y to X being negative. Thus, sentences such as (line 285-286) "All free energy changes shown in Figure 5b are negative, confirming that SUA stabilizes the SAÂ ˚ubase clusters" are absolutely false: negative free energy changes do not in any way confirm "stabilisation". (Also, even when a molecule Y is very strongly bound to a cluster X, this does not in and of itself prove "stabilisation" of the cluster as a whole: it just proves that it is difficult for molecule Y to evaporate. Other constituents of the cluster may evaporate either more or less easily

due to the presence of Y, and this needs to be separately evaluated - just looking at the Y to X binding is not sufficient to claim "stabilization".) I recommend that the authors search the manuscript for all occurrences of the word "stable" in any form (verb or adjective), and rigorously consider whether or not its use is justified. My guess is that almost none of the claims of "stability" or "stabilisation" are actually really justified (apart from perhaps the discussion on hydration, where the effect of H2O concentration is properly accounted for) - the binding of SUA to the clusters tends to be quite weak, with the equilibrium strongly on the side of the reactants even for the highest SUA concentrations claimed by the authors. (See Elm et al., 2017, also cited in the manuscript, for a discussion on what is actually required for a cluster to be "stable" given trace-gas concentration levels of the constituent monomers: this typically means delta-G values far below -10 kcal/mol - which is not reached for any of the SUA addition steps in the manuscript.)

Thanks the reviewer for the clarification of the relationship between cluster stability and $\Delta G$. We agree that a negative value of Gibbs free energy is not sufficient to indicate the stability of a cluster. However, the free energy can imply whether the clustering reaction is favorable or not. Hence, although as reviewer 1 stated, XY is not stable, but a more negative free energy of XY implies the preferable the formation of XY. It means that XY has a high possibility to further react or combine with other species. This is why we used and compared the free energy to judge whether the binding ability of SUA or other precursors.

Furthermore, as the reviewer 1 pointed out, the probability that a core cluster and a precursor molecule can bond together is determined by both the free energy and the precursor atmospheric concentration. Therefore, we examined the possibilities of the core molecule (i.e., SA)/clusters (i.e., SA·base) bonding to SUA under real atmospheric condition (i.e., the ratios for core molecule/clusters bonding to SUA) and how are these bonding ratios for SUA compared to the case for other precursors. To get bonding ratios, first, we have carefully reviewed literatures for reliable atmospheric concentrations of the four precursors studied in this work (i.e., SA, SUA, DMA, AM), and their corresponding concentrations in real atmosphere are listed in Table 3 (please refer to the answer to comment #3 for details). Second, with these reliable atmospheric concentrations, we have calculated the bonding ratios associated with several primary clustering reactions, particularly for SA/SA·base bonding to additional SA or SUA, according to the reviewer 1 illustration. The results are presented in Table 6. Also reported in Table 6 are the interaction energies at 0K ($\Delta H0$), which are indicators of the strengths of hydrogen bonds for individual cluster system.

Based on Table 6, we did some comparisons of thermochemical properties and cluster concentrations at equilibirum between clustering with SUA and with other precursors. We find that, among the clustering reactions between SA and the four precursors (i.e., SA, AM, DMA, and SUA), SA clustering with SUA shows the second lowest free energy (just following SA clustering with DMA). Also, the hydrogen bonding in SA•SUA cluster is as strong as that in SA•DMA and is higher than SA•AM and (SA)2 clusters. Under real atmosphere, SA mostly likely binds to DMA since the bonding ratio for SA to DMA is highest. But if we are looking at the case of SUA to AM, we can see that equilibrium cluster concentration for SA•SUA is almost at the same order as the low bound of that for SA•AM, although that the atmospheric concentration of ammonium is about three to four orders of magnitude of SUA. The chance for SA bonding to SUA is much higher than that for two SA molecules bonding together. When SA•base further clustering with SUA or SA, we can see that the possibility of SA•base bonding to SUA are actually comparable to that bonding to SA. Our calculations of bonding ratios suggest that SUA might be a competitive candidate and

compete with SA to participate in further clustering of SA•base system under real atmospheric condition.

Finally, we summarize these findings and added some discussions to section "3.4 Atmospheric Implication". We also added a description about boding ratio calculations into section 2. "Computational Methods". Followed we also carefully examined and modified all the statements containing "stable", "stability", and "stabilization" in the main text.

**Lines 171-175**

"In addition, the cluster concentration, [cluster], for addition of another monomer to the SA•base dimer with or without SUA is estimated, by considering the atmospheric concentrations of the various precursors,

$$[cluster] = [SA] \times [X]\, e^{\left(\frac{-\Delta G}{RT}\right)} \qquad (9)$$

where the precursor species $X$ corresponds to AM, DMA, or SUA."

**Lines 369-386:**

"In addition, the relative concentrations of the precursor species involved in clustering also govern the cluster distribution in the atmosphere. The estimated cluster concentrations using equation (9) show a cluster concentration (i.e., $10^{-3} \sim 10^2$ cm$^{-3}$) for SUA•SA•DMA (Table 6), suggesting that SUA likely contributes to the further growth of SA•base clusters. The calculated ratios of [SA•X•SUA]/[(SA)$_2$•X] (X denotes AM, DMA, water molecule, or none) under atmospherically relevant concentrations are presented in Table 7. The (SA)$_2$•AM cluster dominates the cluster distribution when SA and SUA concentrations are similar, i.e., [SA•AM•SUA]/[(SA)$_2$•AM] = 1:1000. The estimated cluster ratio of [SA•DMA•SUA]/[(SA)$_2$•DMA] is in the range from 3:100 to 30:1, indicating that SA•DMA•SUA or (SA)$_2$•DMA clusters can be prevalent in the atmosphere. The ratios of [SA•SUA]/[(SA)$_2$] and [SA•W•SUA]/[(SA)$_2$•W] are $10^5$:1 and $10^4$:1, respectively, and hence the SUA-containing clusters are prevalent for both unhydrated and hydrated SA clusters with one water molecule. While sulfuric acid dimer is believed to be an important precursor for NPF (Zhang et al., 2012), our study shows that SUA, which is one of most abundant dicarboxylic acids in atmosphere, inhibits the formation or further growth of sulfuric acid dimer because of its strong interaction with SA in the unhydrated or hydrated SA clusters. Such an inhibiting effect of SUA on the formation or further growth of sulfuric acid dimer is more efficient than ketodiperoxy acid (Elm et al., 2016b)."

**Line 13:**

"Dicarboxylic acids are believed to stabilize pre-nucleation clusters and facilitate new particle formation in the atmosphere…" was changed to "Dicarboxylic acids likely participate in the formation of pre-nucleation clusters to facilitate new particle formation in the atmosphere, …"

**Lines 25-27:**

"the uptake of SUA competes with the uptake of the second SA molecule to stabilize the SA·base clusters at atmospherically relevant concentrations" was changed to "SUA competes with the second SA molecule to cluster with the SA·base clusters at atmospherically relevant concentrations."

**Lines 249:**

We don't have sufficient evidence to prove it and therefore we just removed this statement here:"While the bending of the carbon chain facilitates hydrogen bonding, such bending also induces steric hindrance, which partially cancels out the energy due to additional hydrogen bonding."

**Lines 368-369:**

"All free energy changes shown in Figure 5b are negative, confirming that SUA stabilizes the SA·base clusters." was changed to "The formation of SA•base•SUA by adding a SUA molecule to the SA•base clusters is energetically favorable (Figure 5b), as reflected by large negative free energies."

**Lines 287-289:**

""…the free energy changes for the SA·DMA cluster by SUA addition are more negative than that for the SA·AM cluster, suggesting that SUA more efficiently stabilizes the hydrated SA·DMA clusters than the SA·AM cluster" was changed to "…the free energy changes for the SA·DMA cluster by SUA addition are more negative than that for the SA·AM cluster, suggesting that the clustering between SUA and the SA·DMA cluster core is more favorable than the case of SA·AM cluster core."

**Lines 369:**

Remove the statement here "SUA is more effective than SA to stabilize the SA·base clusters."

**Lines 381-384:**

**"**Sulfuric acid dimer has been recognized as an important precursor for NPF (Zhang et al., 2012), but our study shows that, as one of most prevalent dicarboxylic acids in atmosphere, SUA inhibits the formation or further growth of sulfuric acid dimer because of its strong interaction with in the unhydrated or hydrated SA clusters.**" was changed to**

"While sulfuric acid dimer is believed to be an important precursor for NPF (Zhang et al., 2012), our study shows that SUA, which is one of most abundant dicarboxylic acids in atmosphere, inhibits the formation or further growth of sulfuric acid dimer because of its strong interaction with SA in the unhydrated or hydrated SA clusters.**"**

**Lines 403-405:**

"Addition of SUA to the SA·base cluster systems is energetically favorable at all hydration levels, suggesting that SUA stabilizes the SA·base clusters and their hydrates" was changed to

"Addition of SUA to the SA•base clusters is energetically favorable at all hydration levels, suggesting that SUA likely stabilizes the SA•base clusters and their hydrates.."

**Lines 420-422:**

"Hence, the uptake of SUA competes with the uptake of another SA to stabilize the SA·base clusters, and the presence of SUA hinders the formation and further growth of SA dimer clusters." was changed to "Hence, SUA competes with SA for addition to the SA•base clusters, but the presence of SUA hinders further growth of SA dimer clusters."

**Lines 422:**

Removed "The hydrated SA·DMA·SUA cluster is capable of binding with additional acid molecules, which not only stabilizes the cluster but also promotes its further growth" since we don't have sufficient evidence.

**Table 3. Typical ranges of gas-phase concentrations (molecules cm$^{-3}$) for sulfuric acid, ammonium, dimethylamine, and succinic acid in the atmosphere.**

| Precursors | Sulfuric acid [a] | Ammonium[b] | Dimethylamine[c] | Succinic acid[d] |
|---|---|---|---|---|
| number concentration | $1\times10^5 \sim 1\times10^7$ | $1\times10^9 \sim 1\times10^{11}$ | $1\times10^7 \sim 1\times10^9$ | $1\times10^8 \sim 1\times10^9$ |

[a] Weber et al. (1999), Nieminen et al. (2009), and Zhang et al. (2012).

[b] Seinfeld and Pandis (1998).

[c] Kurten et al. (2008).

[d] Ho et al. (2007).

**Table 6. Gibbs free energy ($\Delta G$, kcal mol$^{-1}$), interaction energy ($\Delta H0$, kcal mol$^{-1}$), and typical cluster concentration at equilibrium for basic clustering reactions. The right-hand side of clustering reactions is the product clusters in equation (9), and the core clusters and addition molecules in equation (9) are listed here as well.**

| Cluster reactions | $\Delta G$ | $\Delta H0$ | Cluster | | [Cluster] (cm$^{-3}$) |
|---|---|---|---|---|---|
| | | | Core cluster | Molecule for addition | |
| SA+SA ↔ (SA)$_2$ | -3.72 | -13.08 | SA | SA | $10^{-7}\sim10^{-3}$ |
| SA+SUA ↔ SA•SUA | -8.61 | -17.94 | SA | SUA | $10^{0}\sim10^{3}$ |
| SA+AM ↔ SA•AM | -6.36 | -14.38 | SA | AM | $10^{-1}\sim10^{-3}$ |
| SA+DMA ↔ SA•DMA | -11.41 | -18.38 | SA | DMA | $10^{1}\sim10^{5}$ |
| | | | | | |
| SA•SUA+SA ↔ (SA)$_2$•SUA | -1.02 | -11.04 | SA•SUA | SA | $10^{-14}\sim10^{-9}$ |
| SA•AM+SA ↔ (SA)$_2$•AM | -9.46 | -19.53 | SA•AM | SA | $10^{-9}\sim10^{-3}$ |
| SA•AM+SUA ↔ SA•AM•SUA | -6.20 | -16.01 | SA•AM | SUA | $10^{-8}\sim10^{-3}$ |
| SA•DMA+SA ↔ (SA)$_2$•DMA | -10.53 | -21.16 | SA•DMA | SA | $10^{-6}\sim10^{0}$ |
| SA•DMA+SUA ↔ SA•DMA•SUA | -9.86 | -19.07 | SA•DMA | SUA | $10^{-3}\sim10^{2}$ |
| | | | | | |
| (SA)$_2$•DMA+SA ↔ (SA)$_3$•DMA | -6.10 | -15.25 | (SA)$_2$•DMA | SA | $10^{-16}\sim10^{-8}$ |
| SA•DMA•SUA+SA ↔ (SA)$_2$•DMA•SUA | -5.13 | -19.07 | SA•DMA•SUA | SA | $10^{-14}\sim10^{-7}$ |

**-Comment #2.**

Related to the previous point, the most interesting part of the data in terms of evaluating the role of SUA on new-particle formation are the energetics for the formation of the clusters shown in Figure 8, i.e. the clusters with two SA molecules. From table 2, it seems that the SA addition free energies to the (SA)(SUA)(DMA)(W)n clusters vary between +2 and -6 kcal/mol, with the value for the unhydrated cluster being -5.14 kcal/mol. This is significantly HIGHER (more positive) than the free energies of SA addition to the (SA)(DMA) cluster or the (SA)2DMA cluster (without any SUA), which are about -18 and -9 kcal/mol (respectively) according to the coupled-cluster (DLPNO-CCSD(T)) study of Myllys et al https://pubs.acs.org/doi/suppl/10.1021/acs.jpca.5b09762), and even more negative according to older DFT - or RICC2- based - studies. (In the absence of data at the exact level used by the authors of this study a direct comparison is impossible - the authors might want to compute at least the unhydrated SA(2)DMA and possibly (SA)3DMA clusters just to check.) Thus, the data presented here indicates - in direct contrast to the manuscript title - that the presence of SUA actually HINDERS sulfuric acid - DMA nucleation rather than enhancing it. I strongly urge the authors to reformulate their title in light of this observation (as well as the comments on stability presented above). Also claims such as SUA "promoting subsequent growth" seem unlikely to be true, if the net effect of SUA is to decrease rather than increase the uptake of further SA molecules.

To justify the point brought up by the reviewer, we did additional computations for unhydrated $(SA)_2(DMA)$ and $(SA)_3(DMA)$ clusters using the same level of theory in this study. Most stable configurations were added into Figure 6 as below. The addition free energy of SA to the (SA)(DMA) cluster or the $(SA)_2(DMA)$ cluster to form $(SA)_2(DMA)$ and $(SA)_3(DMA)$ are -10.5 and -6.1 kcal/mol at the PW91PW91/6-311++G(2d, 2p) level of theory, respectively (see Table 6). These SA addition free energy values are lower than the case of SA addition to the (SA)(SUA)(DMA) (i.e., -5.14 kcal/mol). Therefore, we agree with the reviewer that the presence of SUA actually hinders SA-DMA nucleation rather than enhancing it. As such, we made the revision as follows.

**lines 305-315:**

"The role of SUA in the subsequent growth of the SA•base clusters was examined by comparing the differences in free energies for adding SA to SA•DMA and SA•DMA•SUA. Computations were also performed for the unhydrated $(SA)_2$•DMA, $(SA)_3$•DMA and $(SA)_2$•DMA•SUA clusters (Table 6). The optimized clusters containing more than one SA molecules are depicted in Figure 6. The free energies of adding SA to SA•DMA and to $(SA)_2$•DMA are -10.5 and -6.1 kcal mol[-1], respectively. The free energy for adding SA to SA•DMA•SUA is -5.14 kcal mol[-1], higher than that of SA addition to SA•DMA. However, with hydration (i.e., $(SA)_2$•DMA•SUA•(W)x), the free energies for adding SA to SA•DMA•SUA•(W)x clusters are negative except for the hydration with six water molecules (Table 2), indicating that SA addition to SA•DMA•SUA is also energetically favorable. In addition, addition of SA to SA•DMA•SUA results in formation of strong hydrogen bond (i.e., -19.1 kcal mol[-1]). Therefore, the clusters containing both the base and organic acid (e.g., SA•DMA•SUA) are capable of further binding with acid molecules to promote the subsequent growth."

**Title:**
Based on the update results, we have changed the title to "**Interaction between Dicarboxylic Acid and Sulfuric Acid-Base Clusters**".

[Figure]

Fig. 6. Most stable configurations of (a) unhydrated (SA)2•DMA, (b) (SA)2•DMA, and (c) the hydrated (SA)2•DMA•SUA clusters. The hydration is with 0-6 water molecules (W).

**-Comment #3.** The authors present somewhat misleading concentration ranges and ratios for SUA and SA. The SA value of 1E5 molecules per cm3 corresponds to fairly clean conditions - in polluted environments the SA concentration can easily be a couple orders of magnitude higher. The SUA concentration of 1E7 quoted is from Los Angeles, which I presume corresponds to fairly polluted conditions (with SA certainly exceeding 1E5!) - in clean conditions the SUA concentration is very likely much lower. The total organic acid concentration range of 1E8...1E9 quoted is valid, BUT the majority of these will be simple monocarboxylic acids - dicarboxylics such as SUA will only account for a small fraction of the total. Thus, the [SUA]/[SA] range used in the paper (1 to 10 000) is obtained by combining minimum values for SA with maximum values for SUA - in reality, the ratio may well be below one in most places, and I find it hard to believe that values of 10 000 will be found anywhere in the atmosphere. Using the range 1 to 10 000 is fine for answering the question "could SUA possibly play a role in NPF anywhere in the atmosphere, even in the best case", but then this should be stated openly, instead of implying that the range used is representative for most areas of the world. Again, together with the two previous issues, this indicates that the statement in the manuscript title is exaggerated at best, and false at worst.

In fact, we did claim in the original text that the value of 1E5 molecules/cm3 is the lower limit of the atmosphere SA, so it does correspond to fairly clean conditions, as reviewer 1 pointed out.

However, as the reviewer suggested, since many studies report that the gas-phase concentration of sulfuric acid in atmosphere typically ranges from $1\times10^5 \sim 1\times10^7$ molecules cm$^{-3}$ (Weber et al., 1999; Nieminen et al., 2009; Zhang et al., 2012), we decided to use this concentration range in our calculations to represent the typical atmospheric condition of SA. Regarding the SUA concentration, we misused the particle-phase concentration as the gas-phase concentration for SUA as reviewer 2 pointed out, therefore we did additional literature review for a reliable data source. However, there is limited studies reporting gas-phase concentration of SUA in atmosphere (most related studies only report particle-phase concentration since dicarboxylic acids are normally semi-volatile and tend to partition most portion into particle phase). One available study to report gas-phase concentration is by Limbeck et al. [2001], based on which the atmospheric concentration of SUA in gas phase is set as $1\times10^7$ molecules cm$^{-3}$ (Please refer to the answer to comment #1 of reviewer 2 for details). We have tabulated the typical atmospheric concentrations of four precursors and added it to the manuscript as Table 3.

According to the reviewer 1' comment, with the updated concentrations of SUA and SA, we haved adjusted SUA/SA ratio, ranging from 1 to 10,000 to 0.1 to 100, used to compute the concentration ratios between SUA•SA•X and (SA)2•X Clusters, with X = W, AM, and DMA (Table 6). Of course, in the revised manusript, the extreme condition still is taken into account (such as the condition with the lowest possible level of sulfuric acid but overestimated level of succinic acid), but it is just for testing the role of SUA on the NPF process, and hence it is believed to be reliable.

**Table 7. Concentration Ratios between SUA•SA•X and (SA)2•X Clusters, with X = W, AM, and DMA.**

| SUA/SA | X=(None) | X=W | X=AM | X=DMA |
|--------|----------|----------|----------|----------|
| 0.1:1 | 3.80E+02 | 5.30E+01 | 4.11E-04 | 3.19E-02 |
| 1:1 | 3.80E+03 | 5.30E+02 | 4.11E-03 | 3.19E-01 |
| 10:1 | 3.80E+04 | 5.30E+03 | 4.11E-02 | 3.19E+00 |
| 100:1 | 3.80E+05 | 5.30E+04 | 4.11E-01 | 3.19E+01 |

With the updated Table 7, modifications on the manuscript include the description in section 2. "Computational Methods" and the related discussions in section "3.4 Atmospheric Implication" as below.

**Lines 167-171:**
"We considered a concentration of $10^8$–$10^9$ molecules cm$^{-3}$ for SUA, according to Ho et al. (2007). For comparison, the concentrations of sulfuric acid, ammonia, and dimethylamine in the atmosphere are typically in the range of $10^5 \sim 10^7$, $10^9 \sim 10^{11}$, $10^7 \sim 10^9$ molecules cm$^{-3}$ (Zhang et al., 2012), as listed in Table 3. Hence, the SUA/SA ratio typically ranges from 0.1 to 100."

**Lines 373-381:**
"The calculated ratios of [SA•X•SUA]/[(SA)$_2$•X] (X denotes AM, DMA, water molecule, or none) under atmospherically relevant concentrations are presented in Table 7. The (SA)$_2$•AM cluster dominates the cluster distribution when SA and SUA concentrations are similar, i.e., [SA•AM•SUA]/[(SA)$_2$•AM] = 1:1000. The estimated cluster ratio of [SA•DMA•SUA]/[(SA)$_2$•DMA] is in the range from 3:100 to 30:1, indicating that SA•DMA•SUA or (SA)$_2$•DMA clusters can be prevalent in the atmosphere. The ratios of [SA•SUA]/[(SA)$_2$] and

[SA•W•SUA]/[(SA)$_2$•W] are $10^5$:1 and $10^4$:1, respectively, and hence the SUA-containing clusters are prevalent for both unhydrated and hydrated SA clusters with one water molecule."

**Minor comments:**

**-Comment #4** In the BPMC conformational sampling, what force field was used for the SA, AM, DMA and SUA molecules? (For water TIP3P was apparently used.) Presumably some variant of the AMBER force field, but this should be stated.

According to the reviewer 1's suggestion, we added following description in the manuscript:

**Lines 104-106:**
"We employed the Generalized Amber Force Field (GAFF) for AM, DMA and SUA following Wang et al. (2004; 2006). The force field parameters from Loukonen et al. (2010) were adapted for SUA and bisulfate ion."

**-Comment #5.** -In reaction equations (e.g. 4, 5), arrows (going in both directions to indicate a reversible reaction) should be used rather than equals signs

Thanks for the reviewer 1's comment, done as suggested.

**-Comment #6.** -On line 218, please specify that the proton transfer occurring upon hydration of (SA)(AM)(DMA)(W5) is the SECOND proton transfer, forming the sulfate dianion (overall, it would be good to separate discussion of first and second proton transfers).

We have modified the statement as **(Lines 220-222**)
 "In our study, proton transfer due to hydration occurs in the monohydrate of SA•AM. A second proton transfer also occurs due to hydration, for example, when SA•AM•DMA•(W)$_5$ clusters are hydrated with one more water molecule (Figures 1a and 3a)."

**-Comment #4.** -It's a bit unclear what's being plotted in Figure 5a, apparently the free energy of the reaction X + nH2O => X(H2O)n where X is some core cluster? This could be explicitly specified. Or even better, plot the stepwise hydration energies instead, as the actual hydration in the atmosphere likely involves addition (or removal) of single water molecules, not e.g. 5 molecules at the same time. (Since the stepwise energies are then referred to in discussing the hydrate equilibria, it would be better to plot them in the first place.)

The original Fig. 5a shows free energies of hydration of nH2O, not the stepwise hydration energies. To be clear, we plotted stepwise hydration energies instead as the reviewer suggested. The new figure 5a is as below. Modification was also done in manuscript correspondingly.

**Lines 261-263:**

"The stepwise hydration free energies for the clusters, along with the number of water molecules contained in the clusters, are presented in Figure 5a. The hydration energies are also provided in Table 2."

**Lines 270:**
"Figure 5a shows that the stepwise hydration energies are negative at most hydration degrees…"

[Figure]

**Fig. 5. Stepwise hydration free energies (a) and the relative Gibbs free energy changes due to addition of one SUA molecule to SA•base clusters (b) at T=298.15 K and p=1 atm. The free energy is calculated at the PW91PW91/6-311++G(2d, 2p) level.**

**2. Reviewers #2**

**Review #2 Summary:** This manuscript presents a theoretical study on the interactions between succinic acid (SUA) and sulfuric acid (SA) – ammonia (AM)/dimethylamine (DMA) clusters in presence of water molecules. The application of the results in atmospheric new particle formation (NPF) is discussed. Overall, this study tackles how a multi-component system, which is more realistic in terms of atmospheric NPF, evolves and provides some of the novel insights into the interaction between organic acids and SA-base clusters. On the other hand, the results and their implication could have been presented in a way that is easier to be followed. The authors are advised to address the following concerns before a recommendation can be made.

Thanks the reviewer for insightful comments.

**-Comment #1.** The atmospheric concentrations of NPF precursors are especially important when one wants to discuss the implication of a theoretical calculation. In fact, the authors failed to find a reliable source for the key species that appear in this study. The concentration of SUA is referred from Kawamura and Kaplan, 1987, which actually presents concentrations of particulate SUA and should not be used in a clustering system. The concentration of SA is set at 10^5 molecule cm^(-3), which is at least one order of magnitude lower than many measurement values.

Thanks the reviewer for pointing out the misuse of the particle-phase concentration for (SUA) for calculations of atmospheric-relavent cluster concentration ratio. However, there is few literatures reporting the gas-phase concentrations for dicarboxylic acids, since these acids are belived semi-volatile and primarily partion into the particle phase. Only one available study was by *Limbeck et al.* [2001], which based on in-situ observations has derived particle/gas partitions for several primary dicarboxylic acid species under the atmospheric condition, including succinic acid we are looking at. Based on their's study, the gas-phase concentration of SUA is about 6.7 ng/m$^3$, which corresponds to $3\times10^7$ molecules/cm$^3$. Therefore, we assume a value of $1\times10^7$ molecules/cm$^3$ to represent the gas-phase concentration of SUA in calculation in this work. Please refer to Table 3 in the answer to comment #1 of reviewer 1 for the typical values of atmospheric concentrations.
  Also, the reported concentrations for sulfuric acid in the atmosphere is typically about 10$^5$-10$^7$ molecules cm$^{-3}$ (Weber et al., 1999; Nieminen et al., 2009; Zhang et al.,2012). Therefore, in the revised manuscript, we used the updated concentrations of SUA and SA to re-estimate the SUA/SA ratio range.We have modified the desciription in section 2. "**Computational Methods**", and the related discussions in section 3.4 "**Atmospheric Implication**". Please refer to the answer to comment #3 of reviewer 1 for the modifications of manuscript since the first reviewer have similar concern.

**-Comment #2.** In many cases, the authors compare delta(G) and then conclude that SUA (and/or other molecules) either promote or hinder the growth of clusters. This is fine when the difference between two delta(G) is large. On the other hand, one probably wants to include the concentrations of gaseous precursors and clusters, do the math, and then obtain something like a branching ratio when the difference is small, instead of simply using "promotion or hindrance". Following this point, I would like to see a clear definition of "promotion or hindrance" in the manuscript. Is it a

comparison between the current step of reaction/clustering or a comparison of the further growth of formed cluster from the current step?

Thank the reviewer for the helpful suggestion. We agree with the reviewer that a branching ratio-like parameter can help us to get a clearer result of "promotion or hindrance" from dynamical view. Hence, future works are needed to get the dynamical data such as rate constants, branching ratio. but in this work, we mainly focused on the thermodynamical study. Considering the reviewer's comment and suggestion, we did additional calculations of bonding ratioes for several primary clustering reactions (Table 6). It also can help us get a clearer result of "promotion or hindrance". Please see the answer to coment #1 of reviewer 1 for more detials about bonding ratios. We have added these discussions to main text (**Lines 369-386**).

The "promotion or hindrance" is a comparison of the further growth of formed cluster from the current steps. When we say "promotion or hindrance", we normally say it from energectic perspetive. For example,

**Lines 313-315**: "the clusters containing both the base and organic acid (e.g., SA·DMA·SUA) are capable of further binding with acid molecules to promote their subsequent growth."

Here we conclude that the clusters containing both the base and organic acid promote their subsequent growth because the SA addition free energies to SA·DMA·SUA are negative, showing a potential to bind more SA moelcules.

**-Comment #3.** Given the comprehensive calculation that has been performed, I suppose the authors could suggest a pathway (or multiple possibilities with relative weight for each) on how the complex clusters are formed? e.g., how is (SA)(DMA)(AM)(SUA)(W)6 formed? Will water be added to cluster at the beginning? A summary like this would be welcome, even for a smaller cluster if such a pathway is too complex for a big cluster.

According to the reviewer's helpful suggestion, the summary and the figure about the possible pathways have been added in the revised manuscript (Fig. 9). If we consider the effect of atmospheric concentrations, SA most likely binds to water molecule rather than other precursors (SA, AM, DMA, or SUA), because water vapor is much more abundant than other precursors. For example, the number concentration of water molecules at RH=50% and T=298K is at $10^{17}$ order of magnitude, with a bonding ratio of 0.1, which is much higher than that for other precursors shown in Table 6. Therefore, the disscussions about clustering pathways in main text are mainly carried on from energy perspective.

**Lines 389-392:**
"Fig. 9 depicts the relative stability of cluster formation from the interaction among SUA, SA, base, and W molecules, showing that the SA•DMA cluster is most stable for the dimers and the SA•DMA•SUA or (SA)$_2$•DMA is most stable for the trimers."

[Figure]

**Fig. 9. Possible pathways for formation of the complex clusters.**

**-Minor comments 4.** The reference list should be carefully checked. Some of the references are not in an alphabetical order. Also, I suppose what in Line 75 should be "Xu and Zhang (2012)".

We have revised the reference citing format accordingly.

**Reference**

Ho, K. F., Cao, J. J., Lee, S. C., Kawamura, K., Zhang, R. J. , Chow, J. C., and Watson, J. G.: Dicarboxylic acids, ketocarboxylic acids, and dicarbonyls in the urban atmosphere of China, *J. Geophys. Res.*, *112*, D22S27, https://doi.org/10.1029/2006JD008011, 2007.

Nieminen, T., H. E. Manninen, S. L. Sihto, T. Yli-Juuti, I. R. L. Mauldin, T. Petäjä, I. Riipinen, V. M. Kerminen, and M. Kulmala (2009), Connection of Sulfuric Acid to Atmospheric Nucleation in Boreal Forest, *Environmental Science & Technology*, *43*(13), 4715-4721, doi:10.1021/es803152j.
Weber, R. J., P. H. McMurry, R. L. Mauldin III, D. J. Tanner, F. L. Eisele, A. D. Clarke, and V. N. Kapustin (1999), New Particle Formation in the Remote Troposphere: A Comparison of Observations at Various Sites, *Geophys. Res. Lett.*, *26*(3), 307-310, doi:doi:10.1029/1998GL900308.
Zhang, R., A. Khalizov, L. Wang, M. Hu, and W. Xu (2012), Nucleation and growth of nanoparticles in the atmosphere, *Chem. Rev.*, *112*(3), 1957-2011, doi:10.1021/cr2001756.

---

## Author Response (AR2)

**Manuscript acp-2018-975**

Dear Jingkun:

Thank you for your effort in guiding the initial review for our paper "Interaction between Dicarboxylic Acid and Sulfuric Acid-Base Clusters Enhances New Particle Formation" (acp-2018-975) by Lin et al. We also appreciated the additional comments from both reviewers. Most of the comments by the reviewers were helpful, and we have revised the paper accordingly. Our point-by-point responses to the reviewer's comments are included below.

Thank you again for your assistance.

Sincerely,   Renyi Zhang

**Reviewer #1**
This is a revised manuscript regarding the interaction between succinic acid and clusters consisting of sulfuric acid-ammonia/dimethylamine in the presence of water molecules using quantum chemical calculation methods under the umbrella of atmospheric new particle formation. Although the quality of this manuscript has significantly improved after the revision, a few concerns remain to be addressed.

1. The author lists in the main manuscript and the supplement should match.
Modified as suggested.

2. Since succinic acid is the only dicarboxylic acid that has been studied, the authors are advised to use "succinic acid" instead of "dicarboxylic acid" in the title.
The title has been changed into "Interaction between Succinic Acid and Sulfuric Acid-Base Clusters".

3. Double check the numbers in Table 6, a number of clusters come with concentrations less than 1 cm-3. Although the authors stated that steady-state equilibrium for clusters is rarely established, these numbers are way too low. Extending to the ratios in Table 7, the current number, if true, really suggests that succinic acid pathway is unimportant either.
The calculations have been double-checked as suggested. We have added the following on p. 17, "The estimated cluster concentrations using eq. 9 and the atmospherically relevant concentrations of the precursor species are $10^{-3} \sim 10^2$ cm$^{-3}$ for SA•DMA •SUA and $10^0 \sim 10^3$ cm$^{-3}$ for SA•SUA (Table 6), suggesting that SUA likely contributes to the further growth of the SA and SA•base clusters.". In addition, we have provided more description on the cluster formation and evaporation on p.17 "Under atmospheric conditions, the cluster growth can be represented by a reversible, stepwise kinetic process in a single or multi-component system,

$$...C_{i-1} \underset{k_i^-}{\overset{+A_{i-1},k_{i-1}^+}{\rightleftharpoons}} C_i \underset{k_{i+1}^-}{\overset{+A_i,k_i^+}{\rightleftharpoons}} C_{i+1}...$$
(10)

where $A_{i-1}$ denotes a monomer species to be added to the cluster $C_{i-1}$ at the $(i-1)^{\text{th}}$ step and $k_i^-$ and $k_i^+$ represent the association and decomposition rate constants of the cluster, respectively. Hence, whether cluster $C_i$ grows or decomposes is dependent on the competition between the forward and backward reactions for $C_i$, which are dependent on the rate constant $k_i^+$ and monomer concentration $[A_i]$ and $k_i^-$ (i.e., the thermal stability of $C_i$, respectively (Zhang et al., 2012). The time-dependent concentration of cluster $C_i$ is derived from the equation,

$$\frac{d[C_i]}{dt} = k_{i-1}^+[C_{i-1}][A_{i-1}] - k_i^-[C_i] - k_i^+[C_i][A_i] + k_{i+1}^-[C_{i+1}]$$
(11)

Note that the eq. 9 is the sum of the steady-state expression of eq.11 over all reaction steps". And on p. 18-19 "It should be pointed out that steady-state equilibrium for pre-nucleation clusters is rarely established under atmospheric conditions for each intermediate step (i.e., eq. 11) and the overall reaction (i.e., eq. 9) of the cluster formation, … Hence, the clusters grow (or evaporate) when $k_i^+[A_i]$ is larger (smaller) than $k_i^-$ (eq. 11) … Furthermore, in addition to SUA, there are many other organic acids particularly those with more functionality, i.e., more carboxylic acid groups and the presence of hydroxyl groups, that also likely contribute to aerosol nucleation".

**Reviewer #2**
The authors have made substantial changes to the manuscript in response to comments by myself and another reviewer, and the manuscript is much improved. I still have some, relatively minor, issues with the manuscript, as detailed below. Once these are addressed I can recommend publication in ACP.

While the claims of "stabilisation", "promotion", "enhancement" etc have been toned down, and I especially appreciate the rewriting of the title, there are still a number of sentences that I feel are not really supported by the computed data.

-First, in the last sentence of the abstract, please explicitly spell out the conditions where the claim holds, i.e. replace the last part of the sentence by something like "promotes new particle formation in the atmosphere, at least in conditions where the SUA concentration is significantly larger than the SA concentration".
We have added a phrase "particularly under polluted conditions with high concentration of diverse organic acids."

-On line 259, the authors state that the negative hydration free energies suggest that "hydration stabilises the clusters". This does not really follow, at least not if "stabilisation" is meant to mean stability toward the evaporation of acid or base molecules, not just toward the evaporation of water. Having a strongly negative free energy for the AB + W => ABW reaction may OFTEN correlate with having lower evaporation rates for the hydrated cluster (i.e. ABW => AW + B or ABW => BW may be slower than AB => A + B) but this is not necessarily the case: if the A + W or B + W reactions have even more negative free energies, then hydration will actually INCREASE the evaporation rate of AB, instead of decreasing it. (Examples of this can likely be found even in the authors' own data. Also note that this applies for any number of A, B & W molecules in the cluster, not only the three-molecule example I discuss above). So, to recap: negative hydration free energies just mean that the cluster likes to be hydrated. It doesn't in and of itself imply anything about the more general "stability'" of the cluster, and claiming such an implication is incorrect.
We have changed our statement as suggested "Figure 5a shows that the stepwise hydration energies are negative at most hydration degrees, suggesting that hydration is thermodynamically favorable".

-Line 272: again this "energetically favourable" term: technically true by some definition, but very very misleading when applied to cases where the free energy of the addition reaction is only marginally negative (i.e. not even close to the approximately -10 kcal/mol threshold needed for the evaporation rate not to vastly exceed the formation rate at trace-gas concentrations).
We have removed this statement.

-Line 287: sometimes the described interactions are indeed "synergetic" (should that be "synergistic"?) , but in many cases it seems they seem to rather be "antagonistic", for example when the presence of SUA actually hinders SA-DMA clustering. This fact, that the multi-component interactions can go "both ways" in terms of their effects of clustering, could be noted here.

We have changed our statement as "It is evident that the Gibbs free energy changes of SUA addition to the multi-component clusters are relevant to the hydration degree and the base types".

-Line 293-297: here the authors engage in some sort of "bait and switch" argumentation which is frankly quite intellectually dishonest. First they acknowledge that the free energy of adding SA to SA*DMA*SUA is higher than that of SA addition to SA*DMA. And then they state - as if this somehow invalidated the former conclusion - that "the free energies for adding SA to SA*DMA*SUA*(W)x clusters are negative". Well yes, even the addition of SA to SA*DMA*SUA without water molecules was NEGATIVE - it was just less negative than the competing addition to SA*DMA. The crucial point here is that the SA + SA*DMA*SUA*(W)x reaction free energies (with x = 1…5) are all ONLY VERY SLIGHTLY negative - actually even less negative than the x = 0 case. So in other words the SA molecule will very very rapidly evaporate from the (SA)2*DMA*SUA*(W)x clusters. Again, the "energetically favourable" argument made here may be technically true (by some, fairly irrelevant, definition of favourable), but completely and absolutely misleading: the (SA)2*DMA*SUA*(W)x clusters are actually very very unstable with respect to SA loss, and this should be honestly admitted here!
We have changed our statements as "With hydration (i.e., $(SA)_2 \cdot DMA \cdot SUA \cdot (W)x$), the free energies for adding SA to $SA \cdot DMA \cdot SUA \cdot (W)x$ clusters are negative (Table 2)".
Furthermore, we have added discussions on atmospheric conditions regulating cluster growth or evaporation on p.18 "Clearly, the ability whether a cluster grows to form a nanoparticle is dependent on the competition between the forward reaction by adding a monomer and the backward reaction by losing a monomer (evaporation) at each intermediate step. Hence, the clusters grow (or evaporate) when $k_i^+[A_i]$ is larger (smaller) than $k_i^-$ (eq. 11). While the evaporation rate relies on the thermodynamic stability of the cluster, the forward rate constant is kinetically controlled, dependent on the interaction (i.e., the natural charges and dipole moments) and kinetical energies between the colliding cluster and monomer (Zhang et al., 2012)".

-There seems to be something wrong with equation 9 or how it is defined: if this the [cluster] variable is supposed to be the absolute concentration a cluster with x SA molecules, y W molecules, z AM molecules, n DMA molecules and m SUA molecules, then the expression should read

[cluster] = [SA]^x[W]^y[AM]^z[DMA]^n[SUA]^m exp(-dG/RT)

If the [cluster] variable is instead some relative concentration measure (as implied in some parts of the text), then this needs to be explained/defined better. Since I don't understand exactly what the authors are calculating, I can also not evaluate the arguments on lines 349-351, where the computed values are used to support an argument about SUA "likely contributing to further growth" - however given the issue with the discussion on lines 293-297 described above I'm a bit suspicious about this claim too.

Eq. 9 has been modified as "$[cluster] = [SA]^m \times [AM]^n \times [DMA]^l \times [SUA]^k \times e^{\left(\frac{-\Delta G}{RT}\right)}$ (9) where $\Delta G$ corresponds to the Gibbs free energy change for the reaction: $m$SA + $n$AM + $l$DMA + $k$SUA→ $(SA)_m \cdot (AM)_n \cdot (DMA)_l \cdot (SUA)_k$".

-The Ho et al reference seems to be about the urban atmosphere in China - this is unlikely to be

representative of global dicarboxylic acid concentrations (which are very likely lower). It may well be that such global estimates do not exist - but then the authors should openly admit this, and also acknowledge that the SUA concentrations that they use likely correspond to the upper end of the global range. (This will require some amendments to the discussion on page 17 - also the use of the phrase "typical abundances" on page 19 is not really justified).

We have added statements in Conclusion part p.19-20 "Various organic acids are produced from atmospheric oxidation of volatile organic compounds (VOCs) from biogenic (i.e., pinenes) and anthropogenic sources (i.e., aromatics). Our results indicate that the multi-component molecular interaction involving organic acids, sulfuric acid, and base species promotes NPF in the atmosphere, particularly under polluted environments. The role of different organic acids with distinct functionality in NPF needs to be further assessed. In particular, future studies are necessary to evaluate both the kinetics and thermodynamics of the interactions of organic acids with SA and base species, i.e., the forward and reverse rates as well as the potential energy surfaces for cluster formation, in order to develop physically-based parameterizations of NPF in atmospheric models".

-The authors suggest on line 379 that "the presence of organic acids typically increases the dipole moments of clusters". This may be generally true, however for comparing the competing mechanisms SA*DMA and SA*DMA*SUA, it's not clear that the SUA-containing clusters will have necessarily have higher dipole moments than the corresponding SUA-free clusters. Since the authors actually have a huge amount of cluster data (including dipole moments) perhaps they could present the dipole moments for all the clusters in Table 2, to see if the argument actually holds for the specific systems studied here?

We have added the dipole moment data in Table 6. In addition, we have added a sentence on p. 17 "Furthermore, the dipole moments of SA•DMA•SUA and SA•AM•SUA are 7.4559 and 8.7764, respectively (Table 6), which are the largest among those of the trimers".

-On line 380, the authors again use the (often only very slightly) negative addition energies to imply that "SUA likely stabilises the SA*base clusters". This is problematic in two ways: first, as discussed above, the authors own data indicates that for example the (SA)2*DMA*SUA*(W)x will actually lose SA very rapidly, and second, as discussed above in the context of hydration, even a strongly negative free energy for e.g. AB + SUA => (A)(B)(SUA) does not necessarily mean that the loss rate of A or B is decreased - it can even increase, if the binding of SUA to A or B individually is even stronger. I suggest just removing this whole sentence.

We have removed the sentence as suggested.

-Analogous to the introduction, please specify also in the conclusion that the "promoting" effect the authors claim to have discovered (which I am overall very sceptical about overall given the very high SA loss rates from the only SUA-containing 2-SA clusters studied in the paper) is only important when the SA concentration is relatively low, AND the SUA concentration high (i.e. it's not enough for the conditions to just be "polluted", they have to be "polluted" in a very particular way).

We have added statements in Conclusion part "Various organic acids are produced from atmospheric oxidation of volatile organic compounds (VOCs) from biogenic (i.e., pinenes) and anthropogenic sources (i.e., aromatics). Our results indicate that the multi-component molecular

interaction involving organic acids, sulfuric acid, and base species promotes NPF in the atmosphere, particularly under polluted environments".

Meanwhile, we have added a phrase in the abstract "particularly under polluted conditions with high concentration of diverse organic acids".

-The Kurtén et al 2008 paper (which by the way seems to be missing from the reference list even though it's cited) is not a good primary reference for atmospheric DMA concentrations (it's a purely computational study).

We have removed the citation "Kurtén et al 2008" and added a reference to the atmospheric DMA concentrations in Table 3 "$^c$Ge et al. (2011)."